

# The blue suns of 1831: was the eruption of Ferdinandea, near Sicily, one of the largest volcanic climate forcing events of the nineteenth century?

Christopher Garrison[1], Christopher Kilburn[1], David Smart [1], Stephen Edwards [1].

[1] UCL Hazard Centre, Department of Earth Sciences, University College London, Gower Street, London, WC1E 6BT, UK.
*Correspondence to*: Christopher Garrison (c.garrison@ucl.ac.uk)

**Abstract.** One of the largest climate forcing eruptions of the nineteenth century was, until recently,
believed to have taken place at Babuyan Claro volcano, in the Philippines, in 1831. However, a recent
investigation found no reliable evidence of such an eruption, suggesting that the 1831 eruption must have
taken place elsewhere. A newly compiled dataset of reported observations of a blue, purple and green sun
in August 1831 is here used to reconstruct the transport of a stratospheric aerosol plume from that
eruption. The source of the aerosol plume is identified as the eruption of Ferdinandea, which took place
about 50 km off the south-west coast of Sicily (lat. 37.1° N., long. 12.7° E.), in July and August 1831.
The modest magnitude of this eruption, assigned a Volcanic Explosivity Index (VEI) of 3, has commonly
caused it to be discounted or overlooked when identifying the likely source of the stratospheric sulphate
aerosol in 1831. It is proposed, however, that convective instability in the troposphere contributed to
aerosol reaching the stratosphere and that the aerosol load was enhanced by addition of a sedimentary
sulphur component to the volcanic plume. One of the largest climate forcing volcanic eruptions of the
nineteenth century would thus effectively have been hiding in plain sight, arguably 'lowering the bar' for
the types of eruptions capable of having a substantial climate forcing impact. Prior estimates of the mass
of stratospheric sulphate aerosol responsible for the 1831 Greenland ice-core sulphate deposition peaks
which have assumed a source eruption at a low-latitude site will therefore have been overstated. The
example presented in this paper serves as a useful reminder that VEI values were not intended to be
reliably correlated with eruption sulphur yields unless supplemented with compositional analyses. It also
underlines that eye-witness accounts of historical geophysical events should not be neglected as a source
of valuable scientific data.



# 1    Introduction

Volcanic eruptions that produce sulphate aerosols in the stratosphere are important climate forcing events (Robock, 2000). Ranked in order of the mass of stratospheric sulphate aerosol produced, the most important climate forcing volcanic eruptions of the nineteenth century are: Tambora, in Indonesia, in 1815 (120 Tg); an unidentified eruption in 1809 (59 Tg); Cosegüina, in Nicaragua, in 1835 (40 Tg); Krakatoa, in Indonesia, in 1883 (27 Tg); and another unidentified eruption in 1831 (17 Tg) (Gao *et al*, 2008). The

combination of the 1831 eruption and the 1835 Cosegüina eruption contributed to delaying the end of the Little Ice Age and, hence, the onset of modern anthropogenic warming, until 1850 (Brönnimann *et al*, 2019). Until recently, the 1831 eruption had commonly been assumed to be an eruption of Babuyan Claro, in the Philippines, which had notionally been assigned a Volcanic Explosivity Index (VEI) of 4 (Zielinski, 1995; Global Volcanism Program, 2013; Arfeuille *et al*, 2014; Toohey & Sigl, 2017). However, Garrison

*et al* (2018) found no reliable evidence of such an eruption in 1831, suggesting that the climate forcing eruption must have taken place elsewhere.

Observations of a blue, purple and green sun occurred around the world in August 1831 (Arago, 1832; Kiessling, 1888; Symons *et al*, 1888). The sun is white when viewed from above Earth's atmosphere. At

Earth's surface, however, its observed colour varies due to scattering and absorption by atmospheric gases and aerosols (Bohren & Huffman, 2004). A sufficiently dense aerosol of solid particles or liquid droplets with a radius of about 0.5 μm, and a refractive index of about 1.5 may alter the observed colour of the sun to a pronounced blue, purple or green (La Mer & Kerker, 1953; Penndorf, 1953; Van de Hulst, 1981; Porch, 1989; Horvath *et al*, 1994; Ehlers *et al*, 2014; Wullenweber *et al*, 2021). Such an aerosol is

occasionally produced by a volcanic eruption, for example, the 1880 eruption of Cotopaxi, in Ecuador (Whymper, 1884) or the 1883 eruption of Krakatau, in Indonesia (Symons *et al*, 1888), or a forest fire, for example, the 1950 Chinchaga fire, in Canada (Bull, 1951; Wilson, 1951). For as long as an aerosol maintains these parameters whilst being transported in the atmosphere, it will produce a sequence of observations of a blue, purple or green sun at different dates and places. Consequently, a sequence of

reported observations of a blue, purple or green sun may be used to reconstruct the atmospheric transport of the aerosol responsible, potentially tracing it back to its source (Symons *et al*, 1888).



Here we use a newly compiled dataset of reported observations of a blue, purple and green sun in August 1831 to reconstruct the transport of a stratospheric aerosol plume from the 1831 eruption. We are thus

able to constrain the location of the eruption to a mid-latitude site between 30° N. and 45° N.. Using additional reports where a blue, purple or green sun was not seen, despite active observation, we are also able to constrain its longitude and, hence, identify it as the eruption of Ferdinandea, which took place about 50 km off the south-west coast of Sicily (lat. 37.1° N., long. 12.7° E.) in July and August 1831. The modest magnitude of this eruption, assigned a VEI of 3 (Global Volcanism Program, 2013), *i.e.* ten times

smaller in terms of volume of ejected material than a VEI of 4, has commonly caused it to be discounted or overlooked when identifying the likely source of the stratospheric sulphate aerosol in 1831 (Camuffo & Enzi, 1995; Zielinski, 1995; Robertson *et al*, 2001, Arfeuille *et al*, 2014; Toohey & Sigl, 2017). However, we hypothesize that convective instability in the troposphere contributed to aerosol reaching the stratosphere and that the aerosol load was enhanced by addition of a sedimentary sulphur component

to the volcanic plume.

## 2      Methodology

For simplicity, we use the term 'blue[(+)]' to include the colours blue, purple and green. A literature search

was undertaken to collect as many reported observations of a blue[(+)] sun in 1831 as possible. The observations compiled by Arago (1832), Kiessling (1888) and Symons *et al* (1888) were traced, as far as possible, to their primary sources. Additional primary sources were identified using a combination of 'structured' searches (*e.g.* reviewing obviously relevant national and local newspapers, scientific journals and collections of published accounts of travel and residence) and unstructured searches (*e.g.* keyword

searches of digital archives). A minimum requirement was that a blue[(+)] sun was observed at least once during the day and that the date and place of the observation was recorded. To determine the boundaries of the region in which a blue[(+)] sun was observed, the search also extended to reports recording active observation but without a blue[(+)] sun having been seen ('null' observations). Most of the source materials studied were originally written in western European languages although at least some were originally



written in Arabic, Mandarin and Russian. Online search engines facilitated access to relevant historical materials and translation tools. Supplemental searches of non-digital archives at, for example, the Observatoire de Paris, the Osservatio Astronomico di Palermo and the British Library were also undertaken. In the course of the search, reported observations of other unusual atmospheric optical phenomena in 1831 were noted and collected and will be referred to in this paper as appropriate.


## 3.    Results

### 3.1    Reported observations

Thirty one primary sources reporting observations of a blue[(+)] sun are summarized in Appendix A. The text of three representative reports is reproduced in Table 1. Fifteen of these sources had been identified in Arago (1832), Kiessling (1888) and Symons *et al* (1888) but the remaining sixteen are newly identified here. Seventeen primary sources reporting null observations are also summarized in Appendix B. The original language of the sources in Appendices A and B is English (60%), French (17%), Italian (8%), 100    German (6%), Spanish (4%), Catalan (2%) and Arabic (2%). By type, they comprise newspaper reports (35%), published accounts of travel and residence (25%), observational (meteorological) registers published in newspapers or scientific journals (19%), communications to learned societies published in scientific journals (15%), letters to newspapers (4%) and other published records (2%).

The reported blue[(+)] sun observations took place between 3 August 1831 (source [A1]) and around 28 August 1831 (source [A31]) in Europe, the Caribbean, the north Atlantic, the United States and China (Fig. 1). The sites were located at latitudes between 19° N. and 47° N. although about 85% were restricted to between 30° N. and 45° N. (Fig 1; Fig. 2). The locations of the sites are evidently biased toward regions which had comparatively high population densities in 1831 and, moreover, populations which were likely 110    to report observations in a form which remains accessible today, in particular, in Europe and on the east coast of the U.S.A.. Carpenter (1884) reports that a blue sun was seen "at Washington", in the U.S.A., in "October, especially October 12…[and]…October 13…" 1831. However, this appears to be the result of



| Source No. | Date (Aug.) | Place | Text |
|---|---|---|---|
| A5 | 8 | Palermo, Sicily, Italy. | "Dalle 6 in poi il sole attraversa le dense nebbie presentava un disco con una placida luce bianca turchina; al tramontare lasciò verso ponente una luce rossastra che si prolungò sino a sera avanzata." (Cacciatore, 1831b)<br><br>Translation: From 6 P.M. onward, the sun observed through the dense fog appeared as a pale whitish-blue disc; a reddish light to the west after sunset lasted until late in the evening. (Author's own) |
| A10 | 10 | St Severs, Nouvelle-Aquitaine, France. | "…vers cinq heures du soir…Le soleil était rond et blanc comme une lune, c'est à dire qu'il était dépourvu de rayons apparens, et qu'on pouvait le regarder en face sans que la vue en fût nullement offensée. Une heure après, cet astre était d'un bleu pâle décidé, toujours dépourvu de rayonnement, et l'horizon de son coucher était d'un rouge vif, comme cela s'observe fréquemment dans les journées chaudes. Une sorte de brume éloignée de la terre, et de densité différente, était uniformément répandue dans les régions supérieures, et voilait l'astre du jour…Dans la journée, on avait remarqué que les objets éclairés par les rayons à nu du soleil avaient une teinte bleuâtre." (Dufour, 1831)<br><br>Translation: "…about five o'clock in the afternoon…the sun appeared round and white like a moon; that is to say, it emitted no apparent rays, and could be steadfastly regarded without dazzling or in any manner affecting the eyes. An hour afterwards, it appeared of a pale blue colour, but still destitute of rays; and the horizon, at its setting, was of a deep red, such as is frequently observed after a very hot day. A kind of mist, at a considerable distance from the earth, and of trifling density, was uniformly spread in the upper regions of the atmosphere, and veiled the sun…During the day, the objects exposed to the direct rays of the sun had been observed to assume a blueish tint." (London and Paris Observer, 1831) |
| A22 | 13, 14 | Norfolk, Virginia, U.S.A. | "We have all seen the sun of a dusky red or copper color; but who, until Saturday, the 13th of this month, ever saw it clad in sky blue and pea green? On Saturday, and yesterday morning at its rising, it was of a light but lively green, and as it ascended above the horizon, changed first to cerulean, then to silver white, and finally to pale yellow, when its beams no longer permitted the intrusive gaze of the multitude. And so in its decline, about 5 o'clock in the afternoon it appeared like a globe of silver through the thick haze which overspread the Heavens, shorn of its beams, and gradually assumed the cerulean tint, from which it passed to a light green. A black spot near the centre, was discernible by the naked eye, apparently of the size of a walnut, and with a good spy glass, two others were distinctly visible. In an hour after the sun had set, the horizon in the Northwest exhibited a glare of ruddy light, bearing a strong resemblance to the reflection of a large fire." (Washington National Intelligencer, 1831) |

**Table 1.** Three representative reported observations of a blue[(+)] sun in 1831.




an erroneous conflation between the date and place of an observation on 12 August 1831 and 13 August 1831 in Alexandria, Virginia, about 10 km from Washington (source [A18]), and the publication of this observation in the October edition of Niles' Weekly Register (1831).

Additional observations of blue[(+)] sunlight in China in the summer of 1831 are reported in a Mandarin-language compendium of meteorological records (Zhang, 2004). However, this compendium is itself based on an earlier compendium and the primary sources are not available. These observations have accordingly not been included in the present analysis.


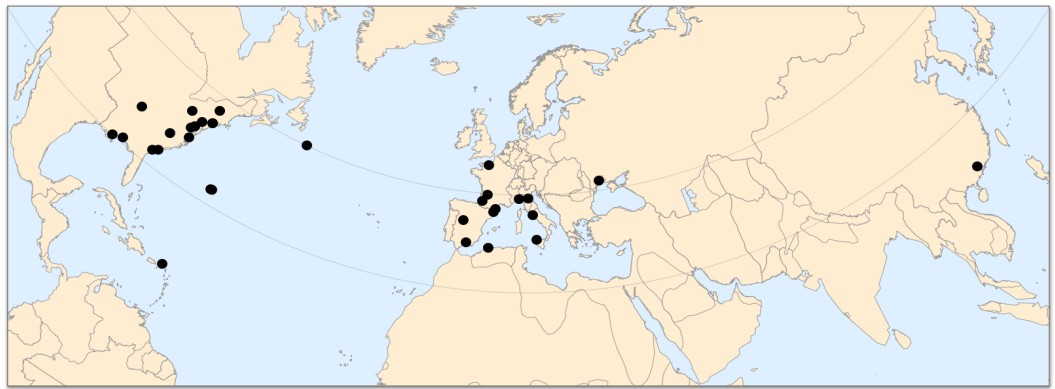

**Figure 1.** Location of blue[(+)] sun observations reported in August 1831 (Appendix A). The latitude band shown extends from 30° N. to 45° N. National borders are shown as correct for 1831 (based on data from Mathematica v. 12.0, Wolfram Research).



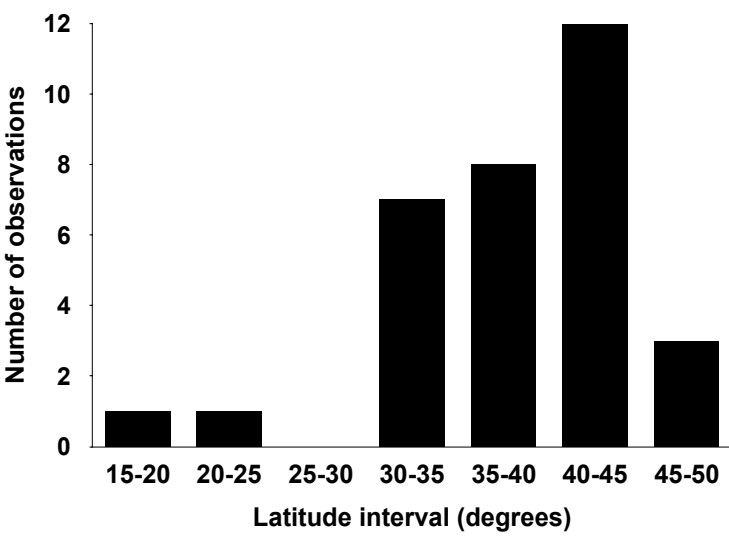


**Figure 2.** Number of blue[(+)] sun observations reported in August 1831 in different latitude bands (Appendix A).

## 3.2    A sequence of observations of a blue[(+)] sun

The reported observations of a blue[(+)] sun in twenty six of the thirty one sources (Appendix A) form a
connected sequence. The earliest observation in the sequence is that reported in Palermo, Sicily, Italy (lat.
38.1 ° N, long. 13.4° E), at 6 P.M. on 8 August 1831 (source [A5]; Fig. 3(a)). Over the following nine
days, the locations of the observations move westward, from Europe to the United States, as well as
spreading to the north and south, largely across the 30° N. to 45° N. latitude band, but occasionally further
north (to about 50° N.; source [A12]) and further south (to about 20° N.; source [A19]) (Figs. 3(b - j) and
4). The observations reported around 28 August 1831 (source [A31]) plausibly extend the connected
sequence further westward to China (Fig. 4).





**Figures 3 (a-j).** Progression of observations of a blue[(+)] sun reported between 8 and 17 August 1831 (Appendix A). The sequence of observations runs day-by-day from 8 August (Fig. 3 (a)) to 12 August (Fig. 3 (e)) and from 13 August (Fig. 3 (f)) to 17 August (Fig. 3 (j)). No reported observations of a blue[(+)] sun have been identified on 16 August (Fig. 3 (i)).





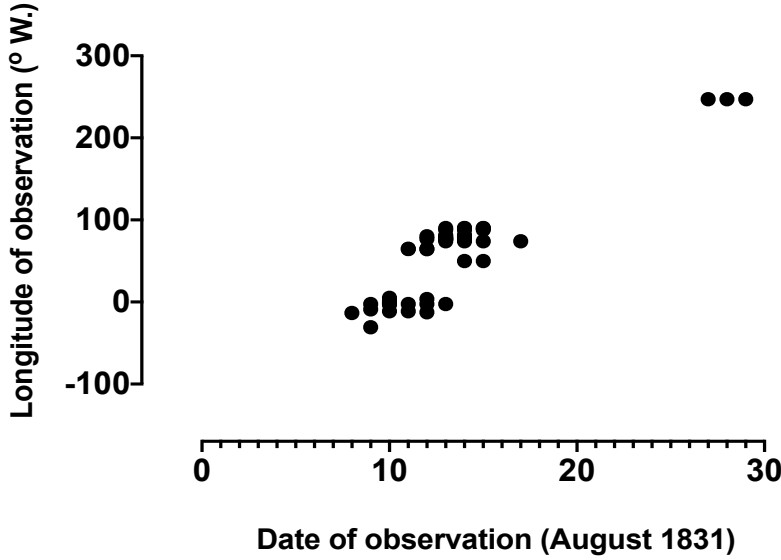

**Figure 4.** Longitude of blue[(+)] sun observations reported in August 1831 (Appendix A).

At locations to the east of Sicily from Malta to India, including those across the 30º N. to 45º N. latitude

band, the sources identified reported null observations, *i.e.* despite observations being actively recorded

no blue[(+)] sun was seen (Appendix B; Fig. 5). The same was true for locations further north and north-

west of Sicily from northern Italy, Switzerland, Hungary and Germany to the United Kingdom (Appendix

B; Fig. 5). No reported observations could be found from locations to the south or south-west, across the

Sahara desert. On this basis, the eastern boundary of the region in which the observation of a blue[(+)] sun

was reported in August 1831 can be delineated approximately with curve A – A' in Fig. 5.






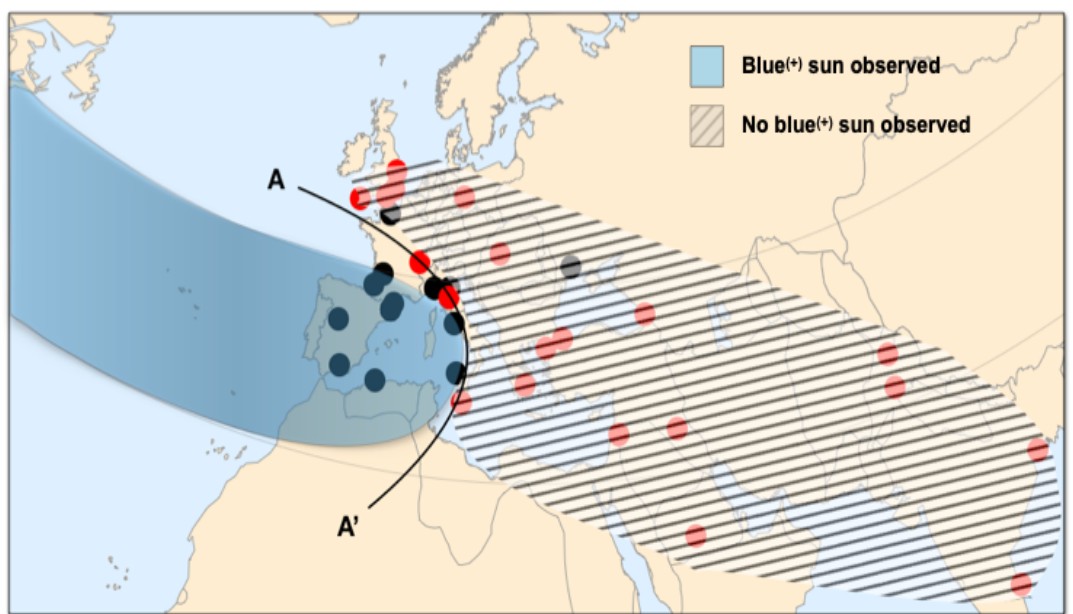

**Figure 5.** The boundary between the region in which observations of a blue[(+)] sun were reported in August 1831 (Appendix A) and the region in which they were not (Appendix B) is delineated approximately with curve A – A'.


### 3.3 Reconstruction of aerosol transport

Given the geographical and temporal distribution of these blue[(+)] sun observations, we propose the following reconstruction of the transport of the aerosol responsible. The aerosol source must have been

located in the vicinity of Sicily, near the intersection of curve A-A' with the centre of the 30° N to 45° N latitude band (Fig. 5). It caused aerosol to be formed between 8 and 13 August 1831 at an altitude with atmospheric circulation from east to west, such that a plume of aerosol lengthened to the west of Sicily (Figs. 6 (a - f)). On 13 August 1831, the aerosol plume extended between about 15° E. and 90° W. in longitude, over an area of about 14,000,000 km$^2$ in the latitude band between 30° N. and 45° N. (Fig. 6

(f); Table 2). Once the source had ceased to cause the formation of aerosol at that altitude, around 14 August 1831, the 'detached' aerosol plume continued to be transported westward (Figs. 6 (g - j) eventually reaching China around 28 August 1831.

**Figures 6 (a – j).** Reconstruction of the generation and transport of the aerosol plume responsible for the blue[(+)] sun observations in August 1831. The sequence of observations runs day-by-day from 8 August (Fig. 6 (a)) to 12 August (Fig. 6 (e)) and from 13 August (Fig. 6 (f)) to 17 August (Fig. 6 (j)). The more minor portion of the aerosol plume which lies outside the 30° N. to 45° N. latitude band is not shown.




## 3.4    Aerosol transport velocity

Calculating the elapsed time between the earliest and latest progressively further westward blue[(+)] sun

observations from 8 to 17 August 1831 in the latitude band $40 \pm 2^o$ N. (Appendix A, col. 7) yields a

transport rate of about $0.97^o$ hour$^{-1}$ for the leading edge of the aerosol plume and about $0.73^o$ hour$^{-1}$ for

its trailing edge (Fig. 7). Taking the mean transport rate of $0.85^o$ hour$^{-1}$ at $40^o$ N. yields a linear velocity

of about 20 m s$^{-1}$.

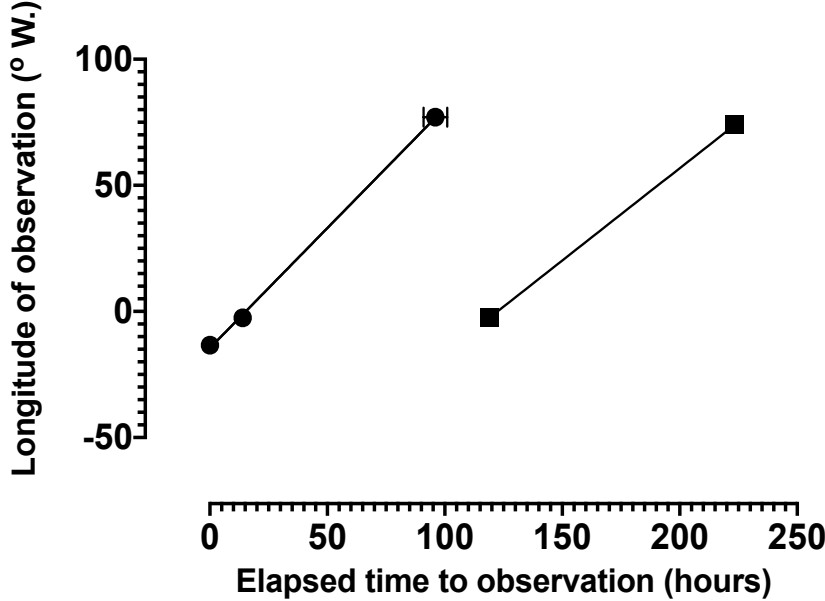


**Figure 7.** Rate of westward transport of the leading (left) and trailing (right) edges of the aerosol plume. Based on

sources [A5], [A6], [A17] and [A18] (left) and sources [A6] and [A30] (right). Elapsed time is measured in hours

from 6 P.M. in Sicily (Time Zone: UTC +1) on 8 August 1831 (source [A5]).






## 3.5 Aerosol transport altitude

The prevailing winds in the mid-latitude upper troposphere in the northern hemisphere are westerly (Barry
& Hall-McKim, 2014). Since the aerosol plume transport took place from east to west, it must therefore
have been transported at an even higher altitude, above the tropopause. The source event type is most
plausibly either a volcanic eruption or a very large forest or bush fire, either of which can inject aerosol
into the stratosphere (Robock, 2000; Khaykin *et al*, 2020); super-eruptions with a VEI of 7 or more may
even inject aerosol into the mesosphere (Costa *et al*, 2018). Given that a super-eruption in the vicinity of
Sicily in August 1831 could not have gone un-recorded, however, it is reasonable to assume that the
aerosol plume must have been transported in the stratosphere.

## 3.6 Aerosol optical depth and mass

In addition to altering the colour of the sun, the scattering and absorption of sunlight by atmospheric gases
and aerosols dims its observed brightness (Bohren & Huffman, 2004). The sun has a visual magnitude M
= -26.74 (Schaefer, 1993). Adapting Stothers (1984a, 1984b), the optical depth ($\tau$) of an atmospheric
aerosol (at zenith angle $z = 0^o$ *i.e.* overhead) is related to the reduction in solar magnitude ($\Delta$M) it
produces at an elevation angle ($\alpha$) as:

$$\tau = \frac{\ln 10}{2.5}(\Delta M \sin \alpha - 0.2) \quad (\alpha > 15^o) \quad [1]$$

where elevation angle ($\alpha$) and zenith angle ($z$) are related as $\alpha = 90 - z$.

Horvath *et al* (1994) report that, due to the physiology of human colour perception, the colour of a blue[(+)]
sun remains too bright to be perceived unless its light has been attenuated by a factor at least $10^{-4}$ and that
it is most readily perceived when its light is attenuated by a factor between $10^{-5}$ and $10^{-6.6}$, equivalent to



a reduction in magnitude between $\Delta M = 12.5$ and $16.5$. Even if too bright to be perceived as such, a blue[(+)] sun will still be sufficiently reduced in magnitude to be able to be viewed with the naked eye without damage or discomfort at a reduction in magnitude between, at the very least, about $\Delta M = 3.4$ (Stothers, 1984a, 1984b) and, more likely, about $\Delta M = 12$ (Schaefer, 1993). At lower reductions in magnitude still, 240   $\Delta M < 3.4$, the sun will be more normal in appearance and too bright to observe with the naked eye.

These three observational phases are illustrated, for example, in two of the reports reproduced in Table 1 (sources [A10] and [A22]). It is noteworthy that the ratio between the brightness of the typical zenithal sun and a white object diffusely reflecting its light is about 80,000 : 1 (Minnaert, 1954) which therefore 245   explains the otherwise paradoxical observation of blue[(+)] sunlight illuminating surfaces and objects when the sun is too bright to be observably blue[(+)] (source [A22]) as the associated attenuation factor of $10^{-4.9}$ is sufficiently close to meet the threshold requirement reported by Horvath *et al* (1994).

Nine of the sources (Appendix A) report the local time at which a blue[(+)] sun was observed. Five of the 250   sources (Appendix A) report the local time at which the sun was observed with the naked eye after having been observably blue[(+)] (in the morning) or before becoming observably blue[(+)] (in the afternoon). The qualitative descriptions of the appearance of the sun in these latter reports, for example, as a 'crystal globe' (source [A8]) or as 'moon-like' (source [A10]), suggest the upper end of the $3.4 < \Delta M < 12$ range in magnitude reduction, *i.e.* $8 < \Delta M < 12$.


Recovering solar elevation angle ($\alpha$) from local time (for example, using the National Oceanic & Atmospheric Administration (NOAA) Solar Calculator, available at: https://gml.noaa.gov/grad/solcalc/) (Appendix A, col. 6) and using the corresponding magnitude range (either $8 < \Delta M < 12$ or $12.5 < \Delta M < 16.5$), expression [1] yields a corresponding instantaneous aerosol optical depth ($\tau$) in each case (Fig. 8). 260   Based on the observations reported in sources [A8] to [A30] between 8 and 17 August 1831, the mean instantaneous optical depth of the aerosol plume is therefore estimated to be $\tau = 4.8 \pm 1.2$ (Fig. 8).

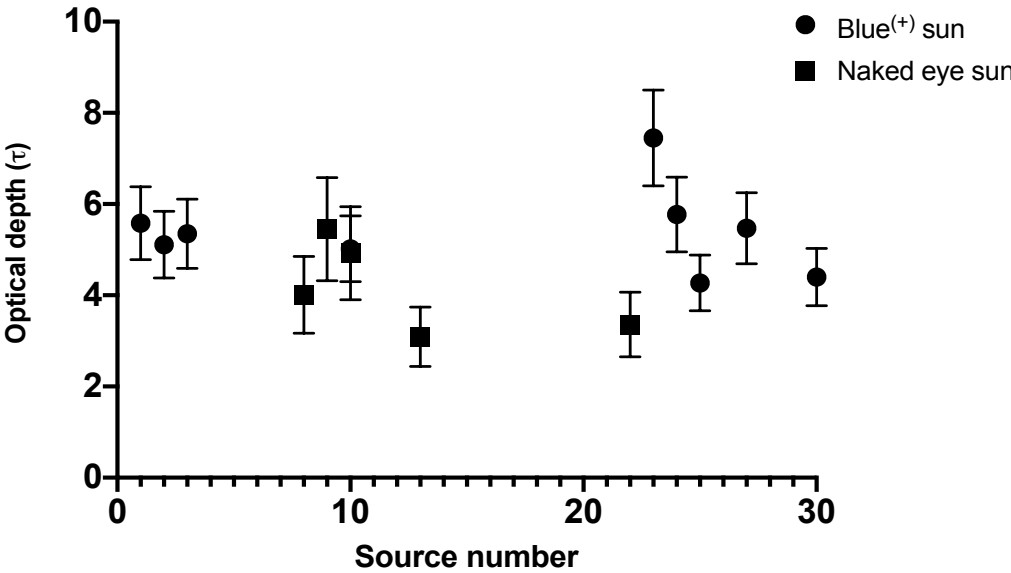


**Figure 8.** Estimated instantaneous aerosol optical depth values. The methodology for estimating instantaneous aerosol optical depth on the basis of the subset of reported observations of a blue[(+)] or naked eye sun which record the local time at which they occurred (Appendix A) is discussed in sect. 3.6.


Had the aerosol plume produced between 8 and 13 August 1831 been homogenous, however, it would have taken six days to pass over each site and reports of six consecutive six days of identical blue[(+)] sun observations would have been expected. In fact, the number of consecutive days of blue[(+)] sun

observations varied between one and five with a mean of about 1.8. The aerosol plume cannot therefore have been homogenous. We instead assume that it was sufficiently dense to produce observations of a blue[(+)] sun, with a mean optical depth $\tau = 4.8 \pm 1.2$, over only a fraction approximately f = 1.8 / 6 = 0.3 of its area. Although the remainder of the aerosol plume was evidently occasionally dense enough to produce ancillary observations of a pale sun (for example, sources [A9], [A13], [A17], [A18]), it will be

neglected in comparison; taking into account background atmospheric Rayleigh scattering, Wullenweber





*et al* (2020) determined that in order for an aerosol which is capable of producing observations of a blue[(+)] sun to do so, it must have an optical depth $\tau > 0.5$.

Adapting Stothers (1984b, 1996), the mass *M* of a homogenous atmospheric aerosol over an area *A* is related to its optical depth ($\tau$) (at z = 0°) as:

$$M = \frac{4\,r\,\rho}{3\,Q}\,\tau\,A \quad [2]$$

where the aerosol particles all have radius r, density $\rho$ and extinction co-efficient Q. To account for the inhomogeneity in this case, we treat the area of the aerosol plume over which it was sufficiently dense to produce observations of a blue[(+)] sun. Using appropriate values for parameters r, $\rho$, Q, $\tau$, A and f (Table 2), expression [2] yields an aerosol plume mass M = 10.1 ± 2.5 x $10^9$ Kg = 10.1 ± 2.5 Tg.

This is a minimum value for the total mass of stratospheric aerosol produced by the source. It does not include the portion of the aerosol plume outside the 30 to 45° N. latitude band, which might be approximately estimated to be 15% (Fig. 2). Further, assuming that the four earlier observations of a blue[(+)] sun in the 30° N. to 45° N. latitude band reported from north Africa on 3 August 1831 (source [A1]) and the north-eastern United States on 4 August 1831 (sources [A2] to [A4]) were caused by aerosol produced by the same source, they would be consistent with two smaller bodies of stratospheric aerosol having been produced slightly earlier during the eruption between 31 July 1831 and 2 August 1831 (sources [A1] to [A3] in Fig. 8).





| Parameter | Approximate value | Rationale |
|---|---|---|
| Extinction efficiency factor (Q) | 2 | As an approximation to Mie's (1908) rigorous description, Van de Hulst (1981) describes the extinction of light of wavelength $\lambda$ by idealised spherically symmetric particles with radius r and (real) refractive index m, in terms of an extinction efficiency function:<br><br>$$Q = 2 - \frac{4}{P}\sin\rho + \frac{4}{P^2}(1 - \cos P)$$<br><br>where $P = (^{4\pi r}/_{\lambda})(m-1)$.<br><br>The conditions required for the observation of a purple sun occur at the first maximum of Q, around P = 4.1, where longer (redder) and shorter (bluer) wavelengths of visible light are extinguished less strongly than intermediate (green) ones; those for a blue sun occur around P = 5.5, where longer (redder) wavelengths of visible light are extinguished more strongly than shorter (bluer) ones; and those for a green sun occur at the first minimum of Q, around P = 7.7, where both longer (redder) and shorter (bluer) wavelengths of visible light extinguished more strongly than intermediate (green) ones (La Mer & Kerker, 1953; Ehlers *et al*, 2014). Assuming a particle refractive index m = 1.5 and that $\lambda$ = 0.55 μm is the centre of the visual light spectrum, the extinction efficiency factor Q varies between 1.5 and 3.2 over this range of parameter P. |
| Refractive index (m) | 1.5 | Aerosol particle refractive index (m) typically varies within a range from one to two, including m = 1.33 (water droplets), m = 1.43 - 1.46 (volcanogenic sulphate droplets), m = 1.46 (organic oil droplets produced by forest fires) and m = 1.55 (desert sand particles) (Penndorf, 1953; Yue *et al*, 1994). |
| Radius (r) | 0.5 μm | Assuming a particle refractive index m = 1.5, the above range of parameter P from 4.1 to 7.7 corresponds to a range of aerosol particle radius from r = 0.36 μm (a purple sun) to r = 0.48 μm (a blue sun) and r = 0.67 μm (a green sun). Non-ideal aerosols produced by natural sources likewise produce the observation of a purple, blue or green sun if their size distribution is dominated by particles in this narrow range (Penndorf, 1953; Horvath et al, 1994), although somewhat broader particle distributions may do so too (Horvath *et al*, 1994; Wullenweber *et al*, 2021). |
| Density (ρ) | 1500 kg m$^{-3}$ | Stothers (1984) assumes a typical aerosol particle density of approximately $\rho$ = 1500 kg m$^{-3}$. |
| Optical depth ($\tau$) | 4.8 ± 1.2 | Sect. 3.6 of this paper. |
| Area (A) | 14,000,000 km$^2$ | Sect. 3.3 of this paper. |
| Homogeneity fraction (f) | 0.3 | Sect. 3.6 of this paper. |

**Table 2**. Parameters for use with expression [2] in sect. 3.6.



### 3.4 Aerosol source


The 1831 eruption of Ferdinandea (also known as 'Campi Flegrei Mar Sicilia' and 'Graham Island') occurred about 50 km off the south-west coast of Sicily (Gemmellaro, 1831; Washington, 1909; Dean, 1980; Global Volcanism Program, 2013). Starting from a submarine base approximately 150 m below sea-level, it produced a volcanic island that first rose above sea-level around 16 July 1831 and

subsequently grew to about 60 m high and 2 km in circumference by the time the eruption ceased around 16 August 1831 (Dean, 1980; Spatola *et al*, 2018). Based on the close co-incidence of place ('in the vicinity of Sicily') and date ('between 31 July 1831 and 13 August 1831'), we therefore identify the Ferdinandea eruption as the source of the stratospheric aerosol that was responsible for the blue[(+)] sun observations in August 1831. It is noteworthy that Riccò (1886) made a similar suggestion in the late

nineteenth century.

The compositional dynamics of volcanic aerosol plumes can be complex (Mather *et al*, 2004). However, volcanogenic aerosol in the stratosphere is typically treated as composed of sulphate droplets containing three-quarters sulphuric acid ($H_2SO_4$) and one-quarter water ($H_2O$) (Zielinski, 1995; Toohey & Sigl,

2017). In order to produce a minimum $10.1 \pm 2.5$ Tg of stratospheric sulphate aerosol, the sulphur yield of the Ferdinandea eruption accordingly could not have been less than about $2.5 \pm 0.6$ Tg.

### 4. Discussion

### 4.1 The plausibility of the Ferdinandea eruption as the aerosol source

The Ferdinandea eruption is described as a small phreatomagmatic ('surtseyan') eruption (Self *et al*, 1989). The remnant cone today lies underwater and has a volume of about 0.06 km[3] (Spatola *et al*, 2018).



The eruption has been assigned a VEI of 3, which is associated with a total volume of erupted tephra of

the order of 0.1 km$^3$ (Global Volcanism Program, 2013). Tephra typically has a density of about $10^{12}$ kg

km$^{-3}$ (Crosweller *et al*, 2012). In order to estimate a sulphur yield for this eruption from these parameters,

the typical ocean island basalt affinity for the Sicily Straits Rift Zone (White *et al*, 2020) is assumed, as

is the pre-eruptive melt sulphur content of 3,000 p.p.m. reported from Etna by Spilliaert *et al* (2006),

which marks the very upper end of the concentration range for this volcanic environment (Oppenheimer

*et al*, 2011). These values yield a maximum sulphur yield of 0.3 Tg, which is about an order of magnitude

smaller than our estimated minimum sulphur yield.

We hypothesize that the release of the additional sulphur was the result of magma interacting with layers

of sulphur-rich sedimentary deposits recently identified at the site of the Ferdinandea eruption (Spatola

*et al*, 2018). These include Messinian evaporites (Spatola *et al*, 2018). Evaporitic sequences are typically

associated with gypsum (CaSO$_4$.2H$_2$O), anhydrite (CaSO$_4$) and halite (NaCl). The 1982 eruption of El

Chichón, in Mexico, injected about 3.8 Tg of sulphur into the stratosphere (Krueger *et al*, 2008), as much

as two orders of magnitude more than would have been expected on the basis of the volume and type of

magma erupted alone (Luhr, 1984; Devine *et al*, 1984; Oppenheimer *et al*, 2011). The two possible

sources for the additional sulphur have been identified as an evaporitic anhydrite bearing sedimentary

layer at the site of the El Chichón eruption and deeper subducted sulphide deposits (Rampino & Self,

1984; Duffield *et al*, 1984; Luhr, 2008). In the case of the Ferdinandea eruption, however, other sulphur-

rich components may be relevant too. In some Sicilian Messinian evaporite contexts where hydrocarbons

are present, sulphate-reducing microbial activity has resulted in conversion of the gypsum or anhydrite to

more easily mobilizable hydrogen sulphide and native sulphur (Ziegenbalg *et al*, 2010). A hydrocarbon

signature has been detected in gas emissions from an active fumerole field about 1 km from the submerged

Ferdinandea cone, although concentrations of sulphur-species were not determined (Coltelli *et al*, 2016).

This hypothesis is supported by eye-witness observations. During the eruption in July and August 1831,

a very strong and unpleasant smell described as a 'stink' of sulphur ('una puzza di zolfo') or of sulphur

and bitumen ('una puzza di zolfo e di bitume') was reported in towns along or near the south-west coast





of Sicily (Gemmellaro, 1831; Russo Ferrugia, 1831). The distance at which it was reported peaked at 100 km on 11 August 1831 (Russo Ferrugia, 1831), *i.e.* during the generation of the reconstructed stratospheric aerosol plume between 8 and 13 August 1831. Silver objects became tarnished at the same time
(Gemmellaro, 1831; Russo Ferrugia, 1831), suggesting that sufficiently high atmospheric concentrations of hydrogen sulphide (and water vapour) reacted with the silver to form a blackened ('tarnished') surface layer of silver sulphide (Inaba, 1996).

A VEI of 3 is associated with 'possible' stratospheric injection rather than the 'definite' stratospheric
injection associated with a VEI of 4 (Newhall & Self, 1982; Global Volcanism Program, 2013). We hypothesize that injection of the volcanic aerosol into the stratosphere by the Ferdinandea eruption was supported by favourable meteorological conditions. The 2018 eruption of Anak Krakatau, an island volcano on the rim of the Krakatau volcano, in Indonesia, was similar in style and magnitude to that of Ferdinandea (Table 3; Figs. 9 (a – b)). It is likewise assigned a VEI of 3 (Global Volcanism Program,
2013). Crucially, sustained phreatomagmatic activity during the eruption produced a column with an updraught in which the vertical velocity was enhanced by convective instability (Prata *et al*, 2020). For six continuous days, a plume of positively buoyant aerosol was thus able to reach an altitude which varied between 16 and 18 km in height above sea level, at times above the local tropopause at $16.8 \pm 0.8$ km (Prata *et al*, 2020).


Anticyclones with subsiding air typically inhibit deep convection in high summer in the central Mediterranean region even though large amounts of convective instability may be present in the mid-troposphere above an inversion (Taszarek *et al*, 2018). The mean height of the tropopause over the south-eastern Mediterranean in the summer is approximately 14 km (Retalis & Cartalis, 1997). We hypothesize
that phreatomagmatic activity during the Ferdinandea eruption was sufficiently sustained, at times, to produce a column with updraughts enhanced by environmental convective instability above any inversion, such that positively buoyant aerosol was able to reach the lower stratosphere, similar to the Anak-Krakatau eruption.




| Eruption | Ferdinandea ('Campi Flegrei Mar Sicilia' and 'Graham Island') | Anak-Krakatau |
|---|---|---|
| Location | In the Straits of Sicily, 50 km off the south-west coast of Sicily, Italy (lat. 37.1º N., long. 12.7º E.) | At sea level in the caldera of the Krakatau volcano, in Indonesia (lat. 6.1º S., long. 105.4º E.) |
| Date | Around 16 July to 16 August 1831 (with a prior submarine phase from around June to 16 July 1831) | 22 December 2018 to 6 January 2019 |
| Predominant eruption style | Surtseyan (phreatomagmatic) | Surtseyan (phreatomagmatic) |
| Erupted volume | $0.06 - 0.1 \text{ km}^3$ | $0.045 \text{ km}^3$ |
| VEI | 3 | 3 |
| Altitude of local tropopause | Approximately 14 km above sea level | $16.8 \pm 0.8$ km above sea level |
| Max. column height | Above the local tropopause (See sect. 3.5 of this paper) | 18 km |

**Table 3.**         Comparison of the eruptions of Ferdinandea and Anak-Krakatau. References: Dean, 1980; Retalis & Cartalis, 1997; Global Volcanism Project, 2013; Spatola *et al*, 2018; Gouthier & Paris, 2019, Prata *et al*, 2020.





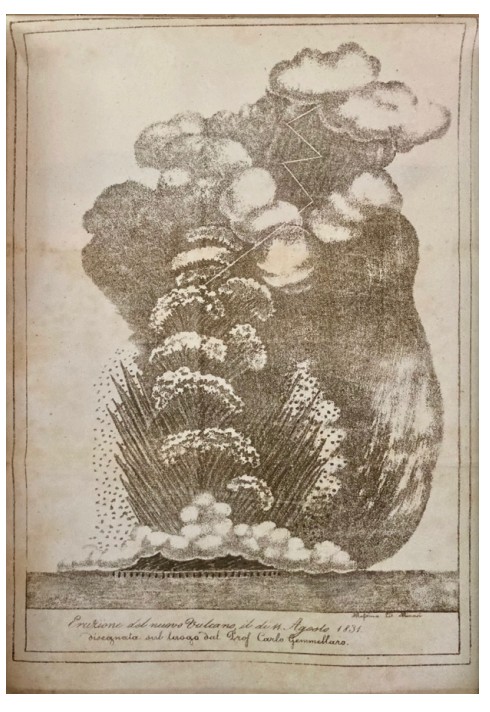

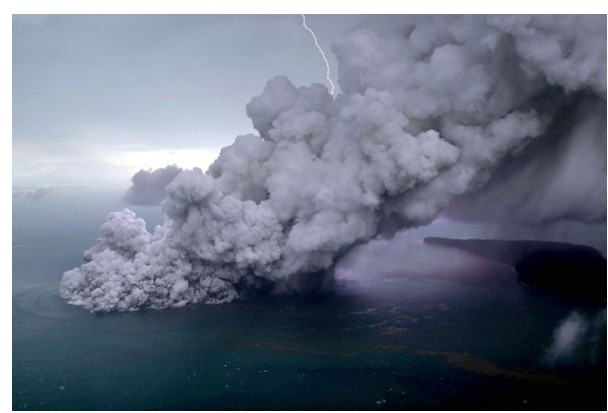

(a)                                                     (b)

**Figure 9 (a).** Sea-level sketch of the Ferdinandea eruption on 11 August 1831 drawn by Gemmellaro (1831). Characteristic features of phreatomagmatic activity are portrayed: pyroclastic material is being explosively ejected in successive cypress-tree or cock's-tail like forms at the base of a rising column of steam and ash (Francis & Oppenheimer, 2004); a 'base surge' is visible. The column appears to be sheared downwind from the observer. A volcanic lightning discharge in the upper part of the eruption column is visible.

**Figure 9 (b).** Photograph of the Anak-Krakatoa eruption taken from a light aircraft on 23 December 2018. Image reproduced with permission: Nurul Hidayat / Antara Foto Agency / Reuters. Characteristic features of phreatomagmatic activity are again visible (Francis & Oppenheimer, 2004) including a 'base surge'. The column is inclined up and to the right of the observer. A volcanic lightning discharge is again visible.




This hypothesis is also supported by eye-witness observations. Approaching the site of the eruption on 5 August 1831, *i.e.* just prior to the generation of the reconstructed stratospheric aerosol plume between 8 and 13 August 1831, Smythe (1831) reported from a distance of 55 km that: "…a stupendous column of white steam was observed, rising majestically far above the western horizon, splendidly illumined (*sic*) by the setting sun…". At that time, on 6 August 1831, continuous episodes of violent 'cypress tree-like' phreatomagmatic explosions were separated by quiescent periods of two to three hours (Smythe, 1831). By 11 August 1831, however, phreatomagmatic activity had significantly intensified: Gemmellaro (1831) reported that the continuous episodes lasted between thirty and forty-five minutes (Fig. 9) and were separated by quiescent periods of only two to three minutes. Stratospheric injection may also have been achieved earlier in the eruption, even if it did not lead to the formation of an aerosol with the parameters necessary to produce observations of a blue[(+)] sun. On 22 July 1831, the eruption column reportedly subtended an angle of at least twenty degrees at a distance of 50 km from the site of the eruption, suggesting that the (visible) column was at least 18 km high (Hoffmann, 1831; Symons *et al*, 1888).

Profiles of temperature, humidity and winds in the atmospheric column in the central Mediterranean in late July and early August 1831 will be required to test the hypothesis that environmental convective instability increased the height of the eruption column, permitting aerosol injection above the tropopause. Although no observations comparable to modern radiosonde ascents exist for this period, a proxy is provided by the recent extension of global atmospheric reanalysis datasets to the early nineteenth century, for example, the Twentieth Century reanalyses (20CR) versions two and three (Compo *et al*, 2011; Slivinski *et al*, 2021). A reanalysis based reconstruction of atmospheric circulation will also permit investigation of whether aerosol transport in different wind directions from the eruption site at different altitudes was responsible for the only blue[(+)] sun otherwise unaccounted for, which reportedly occurred on 9 August 1831 in Odessa, Ukraine (source [A7]; Fig. 5), as well as ancillary reports of unusual haze or fog elsewhere in the Mediterranean or Europe (*e.g.* sources [A1], [A5], [B8], [B13] and [B15]).



## 4.2 Comparison with independent datasets

Assuming that the sulphate aerosol plume was transported in the stratosphere, it could have been expected
to produce occasional observations of a fiery twilight glow from 8 August 1831 onward. Such twilight
glows are often referred to as 'volcanic sunsets' and are characteristic of volcanogenic sulphate aerosols
in the stratosphere (Meinel & Meinel, 1991). The stratospheric altitude is sufficiently high for the sulphate
aerosol to continue to scatter ('reflect') sunlight, so long as it is not blocked by tropospheric clouds, even
when the sun is well below the horizon (Symons *et al*, 1888; Meinel & Meinel, 1991). Stratospheric
aerosol to the east of an observer may therefore produce the observation of a twilight glow before dawn
whereas stratospheric aerosol to the west may produce the observation of a twilight glow after sunset.

Consistent with this expectation, nine of the sources that report observations of a blue[(+)] sun between 8
August 1831 and 17 August 1831 also report observations of a fiery twilight glow whose locations move
from east to west in conjunction with the sites of the blue[(+)] sun observations (Appendix A). In addition,
a longer sequence of twilight glow observations was reported at Palermo between August 1831 and
October 1831 (source [A5]). The sequence appears to have a periodicity of about eighteen days. This
suggests that, although the stratospheric aerosol plume only maintained the parameters necessary to
produce observations of a blue[(+)] sun over about three-quarters of one circuit of Earth (Path A-B-C-D in
Fig. 10), it nevertheless remained dense enough thereafter to continue to produce observations of a
twilight glow over two further circuits of Earth (Path A-B-C-D-A in Fig. 10; Fig. 11). A periodicity of
eighteen days is equivalent to a transport rate of 20° day$^{-1}$ or about 0.84° hour$^{-1}$, very close to the aerosol
plume transport rate (0.85° hour$^{-1}$) estimated in Sect. 3.4 on the basis of the blue[(+)] sun observations.






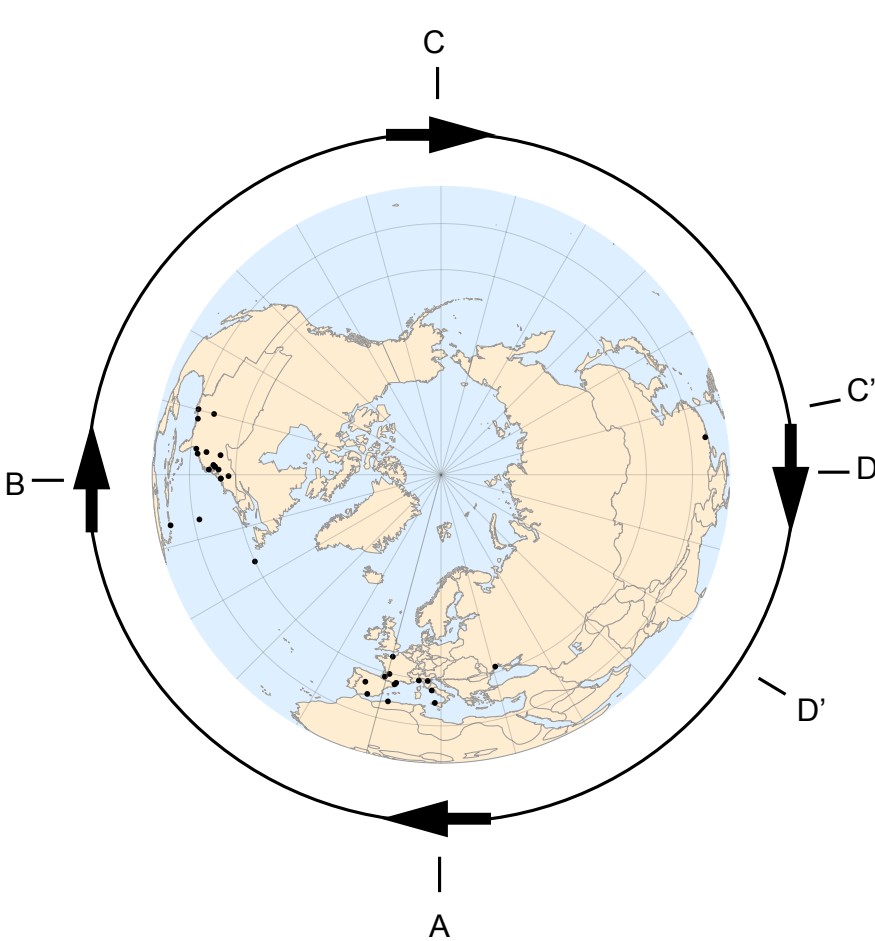

**Figure 10.** Aerosol transport path (A – B – C – D – A) around Earth in 1831. The location of the blue[(+)] sun
observations reported in August 1831 (Appendix A) is shown. The last reported observation of a blue[(+)] sun
occurred around 28 August 1831 in China (source [A31]) near point C' whereas the first observations where no
blue[(+)] sun was reported occurred in India and Pakistan (sources [B1] to [B4]) near point D'.





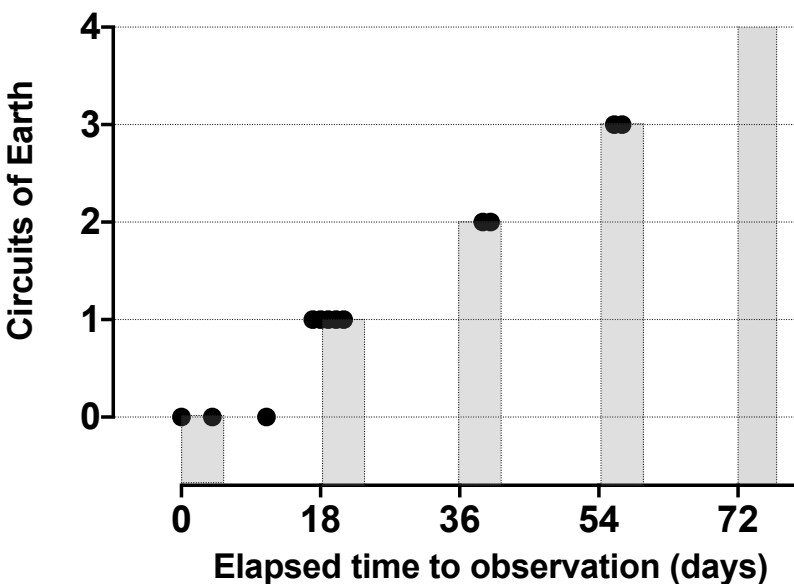

**Figure 11.** Twilight glow observations reported in Palermo, Sicily, Italy between August and October 1831 (source [A5]). Elapsed time is calculated in days from the twilight glow observation on 8 August 1831 (Source [A5]). The periodicity of the observations is consistent with the aerosol responsible completing one circuit of Earth around transport path (A – B – C – D – A) (Fig. 10) in approximately eighteen days.


Six of the null observation sources also report observations of a twilight glow in August 1831 (Appendix B), appearing to reflect the initial transport of the stratospheric aerosol plume or the two smaller precursory bodies of stratospheric aerosol (*e.g.* sources [B12], [B13] and [B14]) and / or the return of the aerosol plume after a first (*e.g.* sources [B5] and [B14]) or second circuit (*e.g.* source [B16]) of Earth.






As the stratospheric sulphate aerosol plume continued to be transported it would have dispersed to form a more homogenous stratospheric sulphate aerosol. Assuming that sulphate aerosol typically has a stratospheric residence time with an *e*-folding timescale (*i.e.* the time taken to decline by a value $e^{-1} = 0.37$) of about one year (Robock, 2000), approximately 63% of it would have returned to the troposphere

after one year and 86% after two years. Assuming that the earliest possible sulphate deposition on the Greenland ice-sheet from the Ferdinandea eruption could have taken place *via* a direct tropospheric route from July or August 1831 and that significant deposition *via* the stratospheric route could continue for no more than two years (Robock, 2000), an increase in sulphate deposition to be detected in Greenland ice-cores between July or August 1831 and around August 1833 would therefore be expected. The magnitude

of the expected increase should correspond to our estimated minimum sulphur yield for the Ferdinandea eruption of about $2.5 \pm 0.6$ Tg.

Consistent with this expectation, Sigl *et al* (2013) report a peak in sulphate deposition in a Greenland ice-core between $1831.4 \pm 0.25$ and $1833.7 \pm 0.25$. Assuming a source eruption of Babuyan Claro, in the

Philippines, Toohey & Sigl (2017) estimate that the sulphur yield of the eruption responsible for the peak was $12.98 \pm 3.41$ Tg. However, if the source eruption had been located at a mid- or high-latitude site, instead of a low-latitude site, this estimate could be reduced by as much as 60% (Toohey & Sigl, 2017). Assuming a source eruption at an unidentified site in the northern hemisphere (and using a different set of Greenland ice-cores), Gao *et al* (2008) estimate the sulphur yield of the eruption responsible for the

peak to have been lower at about 4.2 Tg. Our estimated minimum sulphur yield for the Ferdinandea eruption therefore already represents about 60% of the total eruption sulphur yield estimated by Gao *et al* (2008).

Based on this close co-incidence of expected and actual sulphate deposition profile (and although more

minor contributions from other sources cannot be ruled out), we therefore identify the Ferdinandea eruption as the source of the climate forcing stratospheric sulphate aerosol in 1831.



## 5.     Conclusions


One of the largest climate forcing volcanic eruptions of the nineteenth century took place in 1831 (Zielinski, 1995; Gao *et al*, 2008; Arfeuille *et al*, 2014; Toohey & Sigl, 2017). Here we have used a newly compiled dataset of reported observations of a blue, purple and green sun in August 1831 to reconstruct the transport of a stratospheric aerosol plume from that eruption. We are thus able to constrain the location

of the eruption to a mid-latitude site between 30 º N. and 45º N.. Those prior estimates of the mass of stratospheric sulphate aerosol responsible for the 1831 Greenland ice-core sulphate deposition peaks which assumed a source eruption at a low-latitude site will therefore have been overstated (Zielinski, 1995; Arfeuille *et al*, 2014; Toohey & Sigl, 2017). For a given mass of stratospheric sulphate aerosol, Toohey *et al* (2019) have recently demonstrated that eruptions at mid- or high-latitudes can produce

stronger hemispheric climate forcing than low-latitude eruptions.

Using additional reports where a blue, purple or green sun was not recorded, despite active observation, we are also able to constrain the longitude of the eruption and, hence, are able to identify it as the eruption of Ferdinandea, which took place about 50 km off the south-west coast of Sicily (lat. 37.1º N., long. 12.7º

E.) in July and August 1831. The eruption is assigned a VEI of 3 (Global Volcanism Program, 2013). Its modest magnitude has commonly caused it to be discounted or overlooked when identifying the likely source of the stratospheric sulphate aerosol in 1831 (Camuffo & Enzi, 1995; Zielinski, 1995; Robinson *et al*, 2001, Arfeuille *et al*, 2014; Toohey & Sigl, 2017). However, we argue here that the Ferdinandea eruption must be considered in the context of its geological and meteorological environment, rather than

in isolation. We hypothesize that despite its modest magnitude, its magmatic system was nevertheless sufficient to trigger the release of sulphur from sedimentary deposits at the site of the eruption and that convective instability in the troposphere contributed to aerosol reaching the stratosphere, such that the Ferdinandea eruption did indeed result in the production of the stratospheric sulphate aerosol in 1831.




Our reconstruction of the transport of the stratospheric aerosol plume can be tested and improved with a search for further sources reporting the observation of a blue, purple or green sun in 1831. Our two hypotheses as to the Ferdinandea eruption can be tested through further study of its magmatic system and geological context as well as with a reanalysis based reconstruction of atmospheric circulation in July and August 1831. If we are correct, one of the largest climate forcing volcanic eruptions of the nineteenth

century would thus effectively have been hiding in plain sight. It would arguably 'lower the bar' for the types of eruptions capable of having a substantial climate forcing impact. This example serves as a useful reminder that VEI values were not intended to be reliably correlated with eruption sulphur yields unless supplemented with compositional analyses (Newhall & Self, 1982).

It also underlines that eye-witness accounts of historical geophysical events should not be neglected as a source of valuable scientific data (Guidoboni, 2010; Pyle, 2018). The further before the nineteenth century an event took place, the more difficult it is likely to be to be able to collect primary sources reporting any associated observations of a blue, purple or green sun or of other unusual atmospheric optical phenomena such as twilight glows. Nevertheless, where an adequate collection of primary sources can be collated, it

can likewise be expected to be helpful in terms of, for example, reconstructing the circulatory state of the stratosphere (Hamilton & Sakazaki, 2018) or constraining the latitude, longitude and date of the source event which produced the aerosol responsible (Symons *et al*, 1888).


**Author Contributions**

CG carried out the analysis presented in this study and drafted the manuscript. CK and SE provided guidance and expertise on the volcanological and geological contexts and critically reviewed the

manuscript. DS provided guidance and expertise on the meteorological context (including proposing the pertinence of the convective instability mechanism) and critically reviewed the manuscript.



**Competing interests**

The authors declare that they have no conflict of interest.

**Acknowledgments**

The authors would like to thank: James Lequeux (Observatoire de Paris) for undertaking a search of the papers of François Arago at the Observatoire de Paris; Donatella Randazzo (Osservatorio Astronomico di Palermo) for assistance with a search of the archives of the Osservatorio Astronomico di Palermo; Marc Prohom Duran (Servei Meteorològic de Catalunya) for providing an extract from Quintana i Mari (1938) reproducing the observations of Francesc Bolòs (source [A6]); Julia Rodriguez Sanchez (University College London) for assistance with translation of materials from Catalan; Sabine Rodda for assistance with the translation of materials from German; Kai Deng (University College London) and Zheyu Tian (University College London) for assistance with translation of materials from Mandarin; Colin Graham (Gide Loyrette Nouel) for assistance with translation and historiographical interpretation of materials from Mandarin; Alessandro Aiuppa (Università degli Studi di Palermo), Attilio Sulli (Università degli Studi di Palermo), Sergio Calabrese (Università degli Studi di Palermo) and Walter D'Alessandro (Università degli Studi di Palermo) for valuable discussions regarding the geological context of the 1831 Ferdinandea eruption; and Michael Sigl (Universität Bern) for valuable discussions regarding sulphate deposition peaks in Greenland ice-cores in 1831. The author (CG) is also grateful for the grant of an E A Milne Travelling Fellowship from the Royal Astronomical Society (Burlington House, Piccadilly, London, W1J 0BQ, UK) to support a research visit to Sicily in 2018.









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



## References (Appendices)


Alexandria Gazette: "Meteorological Observations Taken at the Alexandria Museum, Aug. 1831", Friday, 2 September, p.1, 1831a

Alexandria Gazette: "The sun on Saturday evening…", Thursday, 18 August, p.1, 1831b

Allgemeine Zeitung München: "Spanien. Madrid, 18 Aug…", Sonnabend (Samstag), 3 September, No. 246, p. 981, 1831a.

Allgemeine Zeitung München: "Pesth, 12 Aug…", Montag, 22 August, No. 234, p. 936, 1831b.

Arago, F.: Communication to the 22 August 1831 meeting of the Académie des Sciences, in: Journal de Chimie Médicale, de Pharmacie et de Toxicologie, Tome VII, Chez Béchet Jeune, Paris, 1831.

Augsburger Ordinari Postzeitung: "Odessa, den 12. Aug. Im hiesigen Courrier liest man Folgendes...", Sonntag, 4 September, No. 243, 1831.

Basurah, H. M.: Auroras in Arabian Peninsula, J. King Saud. Univ. Sci., Vol. 22, No. 2, 195-200, doi: 10.4197/Sci.22-2.15, 2010.


Bayliffe. B. G., Rawes, J. A.: Extracts from the Journal and Ship's Log of the Hon. Company's Ship Repulse (British Library Ref: L/MAR/B/51F.), http://www.heicshipslogs.co.uk/logs/h022.htm#76, 2016.



Bolòs, F.: Historial meteorològic i agricola de l'estació, 1831, in: Quintana i Marí, A., Assaig Sobre El
Clima D'Olot, Servei Meteorologic De Catalunya, Notes D'Estudi, No. 69, Casa D'Assistència President
Macià, Barcelona, 1938.

Breen, P. H.: Nat Turner's Revolt: Rebellion and Response in Southampton County, Virginia, Ph. D.
dissertation, University of Georgia, Athens, Georgia, 2005.

Burnes, A.: Travels into Bokhara; Being the account of a journey from India to Cabool, Tartary and
Persia; Also, Narrative of a Voyage on the Indus, from the sea to Lahore, John Murray, London, 1834.

Cacciatore, N.: Osservazioni Meteorologiche Fatte Nel Real Osservatorio Di Palermo Nel Mese Di Luglio
Dell'Anno 1831, in: Giornale di Scienze Lettere e Arti per La Sicilia, Tomo XXXV. – Anno IX, Luglio,
Agosto e Settembre, Tipografia del Giornale Letterario, Palermo, 1831a.

Cacciatore, N.: Osservazioni Meteorologiche Fatte Nel Real Osservatorio Di Palermo Nel Mese Di
Agosto Dell'Anno 1831, in: Giornale di Scienze Lettere e Arti per La Sicilia, Tomo XXXV. – Anno IX,
Luglio, Agosto e Settembre, Tipografia del Giornale Letterario, Palermo, 1831b.

Cacciatore, N.: Osservazioni Meteorologiche Fatte Nel Real Osservatorio Di Palermo Nel Mese Di
Settembre Dell'Anno 1831, in: Giornale di Scienze Lettere e Arti per La Sicilia, Tomo XXXVI. – Anno
IX, Ottobre, Novembre e Dicembre, Tipografia del Giornale Letterario, Palermo, 1831c.

Cacciatore, N.: Osservazioni Meteorologiche Fatte Nel Real Osservatorio Di Palermo Nel Mese Di
Ottobre Dell'Anno 1831, in: Giornale di Scienze Lettere e Arti per La Sicilia, Tomo XXXVI. – Anno IX,
Ottobre, Novembre e Dicembre, Tipografia del Giornale Letterario, Palermo, 1831d.

Canton Register: "About a week previously, the sun, for several days…", Thursday, 15 September, Vol.
4, No. 18, 1831.





Colton, C.: Four Years in Great Britain. 1831-1835, Vol. 1, Harper & Brothers, New York, 1835.

Davy, J.: X. Further notice of the new volcano in the Mediterranean, Phil. Trans. R. Soc.,, 122, 251-253,
doi:10.1098/rstl.1832.0011, 1832.

DeKay, J.E.: Sketches of Turkey in 1831 and 1832, J. & J. Harper, New York, 1833.

Du Bois de Montpéreux, F.: Voyage autour du Caucase, chez les Tcherkesses et les Abkhases, en
Colchide, en Géorgie en Arménie et en Crimée, Librairie de Gide, Paris, 1839.

Dufour, L.: Communication to the 22 August 1831 meeting of the Académie des Sciences, in: Journal de
Chimie Médicale, de Pharmacie et de Toxicologie, Tome VII, Chez Béchet Jeune, Paris, 1831.

El Correo: "Provincia de Avila…", Viernes, 26 Agosto, No. 489, p. 4, 1831.

Estrella, C. : Fenómeno Meteorológico, in: El Correo, Miércoles, 24 Agosto, No. 488, p. 3, 1831.

Gazetta di Firenze: "Granducato di Toscana…", Martedi, 16 Agosto, No. 98, p. 3, 1831.

Gazetta di Genova: "Fenomeni celesti…", Sabato, 13 Agosto, No. 65, 1831.

Georgia Messenger: "Solar Phenomenon…", Saturday, 27 August, Vol. 9, No. 25, p. 2, Macon, 1831.

Groves, A.: Journal of a Residence at Baghdad during the years 1830 and 1831, James Nisbet, London,
1832.





Hallowell, B.: Communication to the Washington National Intelligencer, in: Washington National
Intelligencer, Monday, 24 August, 1831.

Harvey, A. W.: Communication to Sir David Brewster, 1839, in: British Association for the Advancement
of Science (BAAS), Report of the Tenth Meeting (Held in Glasgow in August 1840), John Murray,
London, 1841.


Hazard's Register of Pennsylvania: 17 September, No. 194, Vol. 8, p. 192, 1831.

Hess, M.: "I was in the town of Trenton, State of Tennessee…", 1831, in: Roberson, D. M.: Margaret
Daveiss Hess – The First Woman to Plead A Court Case in the U.S., Ansearchin News - Tennessee
Genealogical Magazine, Vol. 47, No. 2., pp. 3-8, 2000.

Horner, G., R., B.: An Account of the Cholera which occurred on board the United States' ship John
Adams, 1831, in: Am. J. Med. Sci., Vol. IX, 257-263, Carey & Lea, Philadelphia, 1831.

Hough, F. B.: Results of a series of meteorological observations, made in obedience to instructions from
the Regents of the University, at sundry academies in the state of New York, from 1826 to 1850 inclusive,
p. xiv, Weed, Parsons and Company, Albany, 1855.

Jacquemont, V.: Letters from India; Describing a journey in the British Dominions of India, Tibet, Lahore
and Cashmere, Vol 1., Edward Churton, London, 1834.

Journal de Genève: "Apparence Singuliere qu'a presente le soleil…", Jeudi, 22 Septembre, p. 168, 1831.

L'Abeille: "Nouvelle-Orleans, 17 Août. L'almanach de la Havane…", Jeudi, 18 Août, p. 1, New Orleans,
970 1831.





Laurent, P.E.: Recollections of a Classical Tour Through Various Parts of Greece, Turkey and Italy, Vol. 2, G. and W .B. Whittaker, London, 1821.

London and Paris Observer: "Arts and Sciences, Meetings of Scientific Bodies…", No. 328, 28 August, p. 558, Paris, 1831

London Morning Post: "New York, Aug. 18 – Solar Phenomenon…", Tuesday, 27 September, 1831.

Mathieu: Communication to the 29 August 1831 meeting of the Académie des Sciences, in: Mémorial Encyclopédique et Progressif des Connaissances Humaines, October, No. 10, 1831.

Mobile Register: "Singular Phenomena…", 17 August 1831, in: American Journal of Science and Arts, January, Vol. XXI, p. 198, 1832.


New York Evening Post: "Pittsburgh (Penn.) Aug 12. Singular appearance…", Wednesday, 17 August, No. 9050, p.2, 1831a.

New York Evening Post: "Atmospheric Phenomenon…", Tuesday, 16 August, No. 9049, p.2, 1831b.


Philosophical Magazine: Meteorological Observations for August 1831, July – December, Vol. X, pp. 318-320, 1831.

Porter, D.: Constantinople and its Environs in a Series of Letters, Harper & Brothers, New York, 1835.


Poussou: Lettre de M. Poussou, supérieur de la mission des Lazaristes à Damas, à M. Etienne, procureur-général de la congregation des Lazaristes, 1831, in: Annales de la Propagation de la Foi, Tome Cinquante-Troisieme, pp. 541 *et seq*, L'Éditeur des Annales, Lyon, 1881.



Reid, W.: Communication to Sir David Brewster, 1839, in: British Association for the Advancement of Science (BAAS), Report of the Tenth Meeting (Held in Glasgow in August 1840), John Murray, London, 1841.

Roulin, F. D.: Communication to the 29 August 1831 meeting of the Académie des Sciences, in: Mémorial
Encyclopédique et Progressif des Connaissances Humaines, October, No. 10, 1831.

Rozet, C. A.: Relation de la Guerre d'Afrique pendant les années 1830 et 1831, Volume 2, Chez Firman, Didot, Frères, Libraires, Paris, 1832.

Rozet, C. A. : Voyage dans la Régence d'Alger, Volume 1, pp. 155-157, Arthus Bertrand, Paris, 1833.

Schwabe, H. F. : Beobachtungen an Sonne, Planeten (Erdatmosphäre) und Kometen während des Jahres 1831, in : Kastner, K. W. G. (ed.), Archiv für die gesammte Naturlehre, Vol. 22, 393-395, Johann Adam Stein, Nürnberg, 1831.
Russo Ferrugia, S.: Storia dell'Isola Ferdinandea, pp. 56-57, Trapani, 1831.

Savannah Republican: "Remarkable state of the Atmosphere…", Thursday, 18 August, Vol. XXVIII, No. 168, p.2, 1831.
Schomburgk, R. H.: The History of Barbados, Longman, Brown, Green and Longmans, London, 1848.

Taylor, T.G.: Result of astronomical observations made at the Honorable the East India Company's Observatory at Madras for the year 1831, Orphan Asylum Press, Madras, 1832
Tizzani, V., Croce, G. M. (ed.): Effemeridi Romane, Volume Primo: 1828-1860, Gangemi Editore, Rome, 2015.



Washington National Intelligencer: "Norfolk, Aug. 15 – We have all seen the sun of a dusky red or copper
color…", 19 August, 1831.

Western Carolinian: "The sun had a very singular appearance…", Monday, 22 August, Vol. XII, No. 585,
p.3, 1831.









**Appendix A.** Observations of a blue, green or purple sun in August 1831.

| No. | Lat. (° N.) | Long. (° E.) | Time zone (UTC) | Brief description of source. | Solar Elevation (°) | Elapsed time (hrs) |
|---|---|---|---|---|---|---|
| A1 | 36.7 (est.) | 1.8 (est.) | +1 | Whilst sailing off the coast of Algeria from Oran to Algiers the French military engineer Antoine Rozet observed a 'clear blue' naked eye sun (with a sunspot) through a 'very remarkable' fog between 7 and 7.15 A.M. on 3 August (Rozet, 1833). He also reported that this fog had appeared at intervals all along the north African coast between 15 July and 15 August and that, at Oran, he had seen a naked eye sun through the fog on several occasions for several minutes at a time (Rozet, 1833).<br><br>N.B. The Dordogne left Oran on 1 August and arrived at Algiers on 4 August (Rozet, 1832). | 25.6<br><br>(7.15 A.M.) | |
| A2 | 42.9 | -74.6 | -5 | A meteorological register taken at Canajoharie (New York State, U.S.A.) relates the observation of a 'pale violet' naked eye sun at 5 P.M. on 4 August (Hough, 1855). | 23.3 | |
| A3 | 40.0 (est.) | -76.3 (est.) | -5 | A traveller on the Susquehanna river (Pennsylvania, U.S.A.) related the observation of a 'violet' naked eye sun through a thin cloud 'overspreading the sky' at 5 P.M. on 4 August to the editors of the Lancaster Miscellany (Hazard's Register, 1832). | 24.5 | |
| A4 | 40.4 | -80.0 | -5 | A report from Pittsburgh (Pennsylvania, U.S.A.), reproduced in the New York Evening Post (1831a), relates the observation of sunlight whose colour 'resembled that of the lilach flowers' (*sic*) on 4 August. The article also relates the observation of unusual twilight phenomena from the first week of August (New York Evening Post, 1831a). | | |
| A5 | 38.1 | 13.4 | +1 | Niccolò Cacciatore, the Italian astronomer and Director of the Palermo Observatory (Sicily, Italy), observed a 'pale whitish-blue' sun through a 'dense' fog at 6 P.M. on 8 August (Cacciatore, 1831b). The full text of this report is reproduced in Table 1. He also reported the observation of dense fogs between 23 and 26 July and 5 and 8 August as well as twilight glows on 4, 6, 8, 12 and 19 August and between 25 and 29 | | 0 |





| | | | | | | |
|---|---|---|---|---|---|---|
| | | | | August, 17 and 18 September and 4 and 5 October (Cacciatore, 1831a, 1831b, 1831c, 1831d). | | |
| A6 | 42.2 | 2.5 | +1 | The Catalan naturalist Francesc Bolòs recorded five or six days of observations of unusually coloured and dimmed suns at Olot (Garrotxa, Catalunya, Spain): on 9 and 10 August, the sun appeared 'white', 'silvery', 'shimmering' and 'moon-like' from its rise till 8 A.M., when it began to produce weak sunlight with a 'purplish' colour, remaining in this state for the remainder of the day; the appearance of the sun on 11 and 12 August was much the same, although it appeared to be briefly 'red' before it became 'white' and began to shine earlier at 7 A.M. and a little less weakly than the previous two days; on 13 August the sun rose with a 'blue' colour (and a sunspot) beginning to shine with a 'blueish' colour at 7 A.M. and remaining like this till 5 P.M. when it dimmed again such that at 6.30 P.M. it looked 'white' and 'moon-like'; the sun was brighter on 14 August and was restored to its normal appearance on 15 August (Bolòs, 1831). He also reported the observation of a twilight glow between 9 and 10 August 1831 (Bolòs, 1831). | | 14 (8 A.M.)<br><br><br>119 (5 P.M.) |
| A7 | 46.5 | 30.7 | +2 | A report from Odessa (Ukraine), reproduced in the German newspaper Augsburger Ordinari Postzeitung (1831), relates the observation of an 'almost violet' naked eye sun (with a sunspot) through an 'almost invisible' fog through the whole afternoon on 9 August. The article also relates the observation of unusual twilight phenomena in the first week of August (Augsburger Ordinari Postzeitung, 1831). | | |
| A8 | 44.4 | 8.9 | +1 | A report in the Italian newspaper Gazetta di Genova (1831) relates that the sun appeared at Genoa (Liguria, Italy) through a 'thin layer of vapour' as a naked eye 'crystal globe' (with a sunspot) at 5 P.M. on 9 August, before turning 'pale red' and then 'violet' in colour. | 27.1 | |
| A9 | 40.6 (est.) | -5.0 (est.) | +1 | A meteorological report from Ávila province (Castile and Léon, Spain) in the Spanish newspaper El Correo (1831) relates that at about 5 P.M. on 9 August, the sun was observed to become as 'pale and white as the moon' and that on 10 August, the sun continued to be pale but with a 'bluish' and subsequently 'whitish' colour.<br><br>N.B. A report from Madrid (Castile and Léon, Spain) dated 18 August, reproduced in the German newspaper Allgemeine Zeitung München (1831a), | 37.7 | |



| | | | | | |
|---|---|---|---|---|---|
| | | | | also relates that an unusual appearance of the sun had recently been observed for several days, with a colour varying between 'blue', 'red' and 'white'. | |
| A10 | 43.8 | -0.6 | +1 | A letter from the French naturalist Léon Dufour was read out to 22 August meeting of the Académie des Sciences relating the observation at St Severs (Nouvelle-Aquitaine, France) of a 'white moon-like' naked eye sun at 5 P.M. which turned 'pale blue' at 6 P.M. (Dufour, 1831). The full text of this report is reproduced in Table 1. | 33.7 (5 P.M.)<br><br>22.9 (6 P.M.) |
| A11 | 44.8 (B)<br><br>42.7 (P) | 0.6 (B)<br><br>2.9 (P) | +1 | The French scientist François Arago informed the 22 August meeting of the Académie des Sciences that, on the basis of letters he had received from Bordeaux (B) and Perpignan (P), the same phenomenon observed at St Severs (source [A10]) was observed throughout southern France (Arago, 1831). | |
| A12 | 49.5 | 0.1 | +1 | The French scientist François Arago informed the 29 August meeting of the Académie des Sciences that the same phenomenon observed at St Severs (source [A10]) had also been seen at Le Havre (Seine-Maritime, France) by M. Mathieu on 10 August (Mathieu, 1831).<br><br>N.B. At the same meeting of the Académie des Sciences, the French naturalist François Désiré Roulin related that it had also been seen to the east as far as Bologna (Emilia-Romagna, Italy), where it lasted for several days (Roulin, 1831). | |
| A13 | 41.9 | 12.5 | +1 | The future Roman Catholic Archbishop Vicenzo Tizzani recorded several days of observations of unusually coloured and dimmed suns at Rome (Lazio, Italy) between 9 and 16 August: a thick fog covering the sky about 17:20 P.M. ('ore 22') on the 9th August caused a 'veiled moon-like' naked eye sun to be seen; the appearance of the sun was the same on 10 August; on 12 August, from about 17:16 P.M. ('ore 22') onwards, a naked eye sun was variously seen through a dense fog to be 'turquoise', 'ashy','yellowish' and 'rosy' in colour; and a 'rosy' naked eye sun was seen at sunset on 16 August (Tizzani & Croce, 2015). He also reported the observation of a twilight glow on 3, 9, 10 and 12 August (Tizzani & Croce, 2015).<br><br>N.B. A system of time-keeping in widespread use in (especially southern) Italy at the time reckoned the twenty four hour period to start at sunset, requiring the | 20.8 (5.20 P.M.)<br><br>20.9 (5.16 P.M.) |



|  |  |  |  | re-setting of clocks and watches according to almanacs of changing sunset times (Laurent, 1821). Sunset on 9 August took place at 19:20 P.M. and on 12 August at 19:16 P.M. (National Oceanic & Atmospheric Administration (NOAA) Solar Calculator, available at: https://gml.noaa.gov/grad/solcalc/). |  |  |
|---|---|---|---|---|---|---|
| A14 | 37.2 | -3.6 | +1 | A letter from Sr. Estrella (1831) to the Spanish newspaper El Correo relates the observation, at Granada (Andalucia, Spain), of a 'blue' sun through a 'thin cloud along the horizon' by the 'tarce' (sic) on 12 August. He also relates the observation of twilight glows there on 10, 11, 12 and 13 August (Estrella 1831). |  |  |
| A15 | 32.3 | -64.8 | -4 | Sir David Brewster read a letter from Augustus Harvey, a doctor in Bermuda, before the tenth meeting of the British Association for the Advancement of Science, relating his memory of observation of 'blue' or 'bluish' sunlight there on 11 and 12 August (Harvey, 1839). |  |  |
| A16 | 32.3 | -64.5 | -4 | Sir David Brewster read a letter from Lt-Col. William Reid, Governor of Bermuda, before the tenth meeting of the British Association for the Advancement of Science, relating that the present collector of customs in Bermuda had been on board a boat fifteen miles east of the island on 11 August and had noticed that the sun was of a 'light green' or 'bluish green' colour (Reid, 1839). |  |  |
| A17 | 32.7 | -80.0 | -5 | An article in the Charleston Courier (South Carolina, U.S.A.), reproduced in the Savannah Republican (1831, p.2), relates the observation of five days of unusually coloured and dimmed suns: on 11 August the sun was 'pale' and 'feeble' throughout the day; on 12 August the sun was the same but with the addition of a 'slight bluish tinge'; on 13 August the sun was the same, if less pronounced; on 14 August, the sunlight at noon was a 'very sensible blue' colour and as dim as during the recent eclipse (12 February) whilst the sun was a 'pale green-blue' and could be observed with the naked eye at a few minutes before 6 P.M.; and on the evening of 15 August the sun had still not recovered 'his usual splendour'. The article also relates the observation of a twilight glow on 12 August 1831 (Savannah Republican, 1831). |  | 96±5 (12 Aug) |
| A18 | 38.8 | -77.0 | -5 | A meteorological register taken at the Alexandria Museum (Alexandria, Virginia, U.S.A.), reproduced |  | 96±5 (12 Aug) |



|  |  |  |  | in the Alexandria Gazette (1831a) which relates the observation of five days of unusually coloured and dimmed suns: on 11 August the sun had a pale and 'silver-like' appearance, between 12 August* and 14 August the sun was alternately 'white', 'brassy', 'green' and 'blue' (with a naked eye sunspot) and it began resuming its normal appearance on 15 August. The notes also relate the observation of a twilight glow as the sun set 'each day' at the same time which resembled the 'light of a great fire' (Alexandria Gazette 1831a). |  |  |
|---|---|---|---|---|---|---|
| A19 | 18.7 | -64.3 | -4 | Whilst surveying around the island of Anegada (British Virgin Islands), the British explorer Sir Robert Schomburgk observed the overcast sky to be a 'threatening' dark bluish colour on 12 August (Schomburgk, 1848). |  |  |
| A20 | 30.0 | -90.1 | -6 | A report from New Orleans (Louisiana, U.S.A.) in the local French language newspaper L'Abeille (1831, p.1) relates that, between 12 or 13 August and 15 August, the sun was observed to set in a 'suspended sea' which dimmed its light and made it a 'blue', 'indigo-blue' or 'greenish' colour. |  |  |
| A21 | 38.3 | -77.5 | -5 | An article in the Fredericksburg Arena (Virginia, U.S.A.), reproduced in the Alexandria Gazette (1831b), relates the observation of a pale blue sun (with a naked eye sunspot) on the evening of 13 and the morning of 14 August. |  |  |
| A22 | 36.9 | -76.3 | -5 | A report from Norfolk (Virginia, U.S.A.), reproduced in the Washington National Intelligencer (1831), relates the observation of a variously 'lively green', 'cerulean', 'silver white' and 'pale yellow' sun (with a naked eye sunspot) on 13 and 14 August. At 5 P.M. on 13 August it appeared like a 'a globe of silver through the thick haze which overspread the Heavens, shorn of its beams'. The full text of this report is reproduced in Table 1. The report also relates a twilight glow on 13 August (Washington National Intelligencer 1831). | 22.7 |  |
| A23 | 38.8 | -77.0 | -5 | A letter from the American amateur scientist Benjamin Hallowell (1831) to the Washington National Intelligencer relates that, at about mid-day on 13 August in Alexandria (Virginia, U.S.A.), he observed that the sun shining through a 'body of vapor suspended in the heavens' had a 'silvery' appearance, changing between 3 P.M. and 4 P.M. to a 'greenish- | 34.9 (4 P.M.) |  |



| | | | | blue' and that it descended 'below the body of vapor' about fifteen to twenty minutes before sunset; he noted the presence of a naked eye sunspot. He also reported a twilight glow on 12 August (Hallowell, 1831).<br><br>N.B. Breen (2005) reproduces a similar account from a letter written by Emma Mordecai who relates the observation of a blue sun at about 4 P.M. on 13 August nearby in Richmond (Virginia, U.S.A.). | | |
|---|---|---|---|---|---|---|
| A24 | 32.1 | -81.1 | -5 | A report in the Savannah Georgian (Georgia, U.S.A.), reproduced in the Georgia Messenger (1831), relates that on 13 August a 'blue' naked eye sun was seen from 5 P.M. till sunset and that, although less dim, a 'blue' sun (with a sunspot) continued to be seen on 14 August. | 26.5 | |
| A25 | 30.7 | -88.0 | -6 | A report reproduced from the Mobile Register (Alabama, U.S.A.) (1831) relates that a variously 'pale blue', 'violet' or 'sea-green' naked eye sun (with a sunspot) was seen from 5 P.M. till 6 P.M. on 13 August, a 'bluish' sun was seen on the morning of 14 August and a 'pale green' sun was seen at 6 A.M. on the morning of 15 August (Mobile Register, 1831). | 19.5 (5 P.M.) | 126 (6 P.M.) |
| A26 | 40.7 | -74.0 | -5 | A report published in the New York Evening Post (New York State, U.S.A.) on 16 August relates the observation of a sun for 'several days past' (*i.e.* between 13 and 15 August?) which on rising was 'dull white, slightly tinged with green' and which, between 30 and 45 minutes later, was brighter but with sunlight of a 'faint silvery hue, somewhat greenish, not unlike the color of the silk of green corn' (New York Evening Post, 1831b, p.2). The report also relates the observation of a twilight glow (New York Evening Post, 1831b). | | |
| A27 | 45.0 (est.) | -50.0 (est.) | -3<br><br>or<br><br>-4 | Whilst sailing across the north Atlantic from New York to Liverpool, the American clergyman Calvin Colton observed two or three days of an unusually coloured and dimmed sun: on 14 August, a 'dark purple' naked eye sun (with a sunspot) was seen at around 5 P.M. (although a member of the ship's crew indicated that the phenomenon had begun around 3 P.M. with the 'unusual symptoms' gradually increasing); on 15 August the appearance of the sun was the same although even darker in the afternoon; and it was 'not till the third or fourth day that the heavens began to wear their natural appearances' (Colton, 1835). | 25.1 (5 P.M.) | |



| | | | | | | |
|---|---|---|---|---|---|---|
| | | | | N.B. The Silas Richards left New York on 9 August and arrived at Liverpool on 28 August; Colton believed that the ship was 'on the Banks of Newfoundland, or in the neighbourhood' when the observations took place (Colton, 1835). | | |
| A28 | 35.7 | -80.5 | -5 | A report in the Western Carolinian (Salisbury, North Carolina, U.S.A.) relates the observation of a 'blue' naked eye sun (with a sunspot) on 14 August; the report further relates that 'some of the old inhabitants of this place say that it presented the same appearance in 1816 or 1817' (Western Carolinian, 1831). | | |
| A29 | 36.0 | -90.0 | -6 | A handwritten note inscribed in a family Bible by Margaret Hess relates that she observed the sun to have a 'clear blue culler' (*sic*) in Trenton (Tennessee, U.S.A.) on the afternoon of 14 and the morning of 15 August (Hess, 1831). | | |
| A30 | 40.7 | -74.0 | -5 | A report in the New York Commercial Advertiser (New York State, U.S.A.), reproduced in the London Morning Post (1831), relates that a naked eye sun which was 'green, as the sea water or Brazilian emerald' was observed between 5 P.M. and 7.30 P.M. on 17 August. | 20.1 (5 P.M.) | 223.5 (7.30 P.M.) |
| A31 | 23.1 | 113.3 | +8 | A report in the English language Canton Register newspaper (Guangzhou, China) relates the observation of two parhelia 'here' on 4 September as well as the observation, 'about a week previously' and 'for several days' (*i.e.* between 27 and 29 August?), of a pale green sun both on rising and setting (Canton Register, 1831). | | |







## Appendix B. Null Observations.

| No. | Lat. (º N.) | Long. (º E.) | Brief description of source. |
| --- | --- | --- | --- |
| B1 | 22.0 (est.) | 88.0 (est.) | The Repulse, an Honorable East India Company vessel, was at anchor at the entrance to the Hooghly river (now Bhāgirathi-Hooghly river), West Bengal, India from 5 June till 8 August 1831, before sailing to Penang, Malaysia, and reaching Singapore by 28 August (Bayliffe & Rawes, 2016). Whilst recording daily weather conditions, no observations of Unusual Atmospheric Optical Phenomena (UAOP) were reported in the log-book in July or August. |
| B2 | 13.1 | 80.3 | Astronomical observations were recorded at the Honorable East India Company Observatory at Madras (now Chennai), India, in 1831 (Taylor, 1832). No episodes of unusual stellar dimming were reported. No equivalent meteorological observation book can be found. |
| B3 | 34.0 (est.) | 75.0 (est.) | The French naturalist Victor Jacquemont was travelling in 'Cashmere' (now Jammu and Kashmir, India) between 8 May and 19 September 1831 (Jacquemont, 1834). He reported no UAOP in July or August. |
| B4 | 31.5 | 74.3 | The British explorer Sir Alexander Burnes was resident in Lahore, Pakistan, between 18 June and 16 August before travelling onward to reach Simla (now Shimla), India, in September 1831 (Burnes, 1834). He reported no UAOP in July or August. |
| B5 | 25.0 (est.) | 45.0 (est.) | A record of historical events (Enwan Al-Majed fi Tarekh Najd) in Najd Province, Saudi Arabia, reports unusual twilight phenomena during the first and second week of August 1831 and a twilight glow in the evening of 23 August (Basurah 2010). Two less precisely dated observations of a twilight glow are also related, one lasting three days (between 1831 and 1832) and another which lasted some months (between 1832 and 1833) (Basurah, 2010). |
| B6 | 33.3 | 44.4 | An English Protestant Missionary, Anthony Groves, kept a day-to-day journal of his residency in Baghdad, Iraq, in 1830 and 1831 (Groves, 1832). He reported no UAOP in July or August. |
| B7 | 42.3 | 41.7 | The Swiss naturalist and antiquarian Frédéric Du Bois de Montpéreux, travelled in the Caucasus between 1831 and 1834. Despite, for example, recording weather observations in Redoute-Kalé (now Kulevi, Georgia) on 2, 3 and 4 August 1831, he reported no UAOP in July or August (Du Bois de Montpéreux, 1839). |
| B8 | 33.5 | 36.3 | The head of the Lazaristes mission in Damascus (Syria), M. Poussou, reported in a letter to M. Etienne, Procureur Général of the Lazaristes Congregation, dated 12 September 1831, that for 'about the last two months' (i.e. since about mid-July?) the atmosphere there had been 'laden [with vapours]' and the sun had been pale, including appearing as if seen through 'crêpe' [fabric] for at least fifteen minutes after sunrise' (Poussou, 1831). He also reported the observation of a twilight glow 'during this period', both before sunrise and after sunset, although he does not say whether continuously or occasionally and, if the latter, at which dates (Poussou, 1831). |



| B9 | 41.0 | 29.0 | Gustavus Richard Brown Horner, Surgeon aboard the American vessel, the John Adams, arrived at Constantinople (now Istanbul, Turkey) on 10 August 1831 (Horner, 1831). The John Adams had arrived off the Dardanelles (now Çanakkale Boğazı, in Turkey) on 4 August (see source [B10]). He kept a day-to-day record of the weather. With the exception of an 'atmosphere loaded with vapours impenetrable to the sun' on 22 August, he mentioned no UAOP in July or August (Horner, 1831). |
|---|---|---|---|
| B10 | 40.3 | 26.4 | The American naval officer David Porter arrived off the Dardanelles (now Çanakkale Boğazı, in Turkey) on 4 August 1831 aboard the American vessel the John Adams and reached Constantinople (now Istanbul, Turkey) by 10 August (Porter, 1835). The purpose of his visit to Constantinople was to ratify the first treaty agreed between the United States and the Ottoman Empire. He reported no UAOP in July or August although he did describe a remarkable hail-storm which took place in Istanbul on the same day as the ratification ceremony. |
| B11 | 37.5 | 23.4 | The American zoologist, James DeKay, sailed through the Mediterranean to Turkey in 1831. He gave a close description of an early phase of the eruption of Ferdinandea, off the coast of Sicily, Italy, as he passed it on the morning of the 16 July (DeKay, 1833). The vessel reached the island of Crete, Greece, around 27 July, and then made its way via Milos, Hydra (around 30 July), Poros (a few days before 13 August), Tinos, Andros, Chios, Lemnos, Tenedos and the Dardanelles (now Çanakkale Boğazı, in Turkey) to reach Constantinople (now Istanbul, Turkey) later in August (DeKay, 1833). He reported no UAOP in July or August although he did describe a remarkable hail-storm which took place in Istanbul (see source [B10]) |
| B12 | 35.9 | 14.4 | The British doctor (and Assistant Inspector of Army Hospitals) John Davy reported the observation in Malta of a twilight glow for many evenings successively in August 1831 (Davy, 1832). |
| B13 | 43.8 | 11.3 | An article in the Italian newspaper, the Gazetta di Firenze (Florence, Tuscany, Italy) (1831) reports the observation of a dense haze and a twilight glow in the evenings for several days prior to 13 August. |
| B14 | 46.2 | 6.1 | Referring to the observations of a blue sun made by Léon Dufour in France on the 10th August 1831 (source [A10]), an article in the Swiss newspaper, the Journal de Genève (Geneva, Switzerland) (1831) reports the observation of a twilight glow there in the evenings of 10 and 30 August. |
| B15 | 47.5 | 19.0 | An article in the German newspaper Allgemeine Zeitung München (1831b, p. 936) relates that the air in Budapest, Hungary, was filled with a thin fog on 11 August 1831 which dimmed the distant mountains and 'almost made it difficult for anyone to breathe'. |
| B16 | 51.8 | 12.3 | The German Astronomer Samuel Heinrich Schwabe, whose dedicated solar observations between 1826 and 1843 led to his discovery of the 11-year sunspot cycle, lived in Dessau, Germany. Whilst reporting all of the most notable astronomical and meteorological phenomena he observed in 1831, he mentioned a twilight glow that had occurred on 25, 26 and 27 September (Schwabe 1831). |
| B17 | 50.8 (G) 51.5 (C) | 1.1 (G) 0.3 (C) | Meteorological observations across the United Kingdom for August 1831 were reported by 'Dr. BURNEY' in Gosport (G), 'Mr. THOMPSON' in Chiswick, London (C), 'Mr. GIDDY' in Penzance (P) and 'Mr. VEALL' in Boston (B), all in England, United Kingdom (Philosophical Magazine 1831). No UAOP were reported in the accompanying day-to-day comments. |





| | 50.1 (P) | 5.5 (P) | |
|---|---|---|---|
| | 53.0 (B) | 0.0 (B) | |





