# Peer review of "The blue suns of 1831: was the eruption of Ferdinandea, near Sicily, one of the largest volcanic climate forcing events of the nineteenth century?"

_Climate of the Past, 2021_

## Author Comment (AC1)

Authors' response to Anonymous Reviewer #1

General comments:

This is a well written and interesting study on the origin of numerous observations of blue/green suns in 1831. There will certainly always be some uncertainty, but in my opinion this is a convincing study and according to the evidence presented in the manuscript it seems indeed highly plausible that the 1831 eruption was the one of Ferdinandea. The paper is well suited for Climate of the Past and I recommend publishing the article after considering the following (mainly minor) comments.

**[R1,1]   Thank you. We are grateful for your helpful and detailed comments.**

Apart from the specific comments below it would be good if the paper would also briefly address the following points:

- Were there reports of wildfires in Europe in August 1931?

**[R1,2]   Although it was not a primary focus of the literature search, no reports of very large forest or bush fires in southern Europe in July or August 1831 were identified.**

How does the time lag between the eruption and the occurrence of colored suns differ between Ferdinandea and other eruptions, e.g. Krakatoa?

**[R1,3]   Please see response 2 to Reviewer #3.**

Specific comments:

Lines 77 – 80: Perhaps you can here already refer to the appendices. When reading these sentences I was asking myself: How many newspapers and journals were searched and which ones? This information is provided later, but it would also fit here.

Line 122: "These observations have accordingly not been included in the present analysis."

Probably because Fig. 1 is right below this sentence I was asking myself, whether the point in China refers to the observations of this Mandarin compendium (which is not the case, as I learned later).

Figure 5: what is the meaning of the color of the circles. This is not explained as far as I can tell. In particular: What is the black circle in the "no blue sun" area?

**[R1,4]   We had indeed neglected to specify the meaning of the colour of the points in Figure 5. We have revised the caption to read:**

> **"Figure 5. Locations of observations of a blue[(+)] sun (black points) and of null observations (red points) reported in August 1831 (see Appendices A and B). The boundary between the region where a blue[(+)] sun was observed and the region where it was not is delineated approximately with curve A – A'."**

**The apparently exceptional case of the single black point in the 'no blue[(+)] sun' area represents observation [A7], reported from Odessa on 9 August 1831. As discussed at**

**manuscript lines 441** *et seq*, **it may be that this observation represents the transport of aerosol from the eruption site at a different altitude and in a different wind direction from the others in the connected sequence.**

Fig. 6: I think this Figure is appropriate for this manuscript, but I would add a disclaimer that this is only a very crude depiction of the plume extent and motion.

**[R1,5]   We have revised the manuscript at lines 174 and 186 (Figure 6) so that the word 'reconstruction' is qualified with the word 'approximate'.**

Line 194: "yields a transport rate of about 0.97 deg hour^-1"

This refers to longitude, right? I suggest mentioning this explicitly to avoid confusion.

**[R1,6]   The y-axis in Figure 7 does specify degrees longitude but, to avoid any doubt, we have revised the manuscript at lines 195 and 196, as well as at lines 467 and 468 so that the term '(long.)' is added between the terms 'º' and 'hour$^{-1}$'.**

Section 3.5: I would add that zonal winds in the stratosphere at mid-latitudes depend on the seasons and are easterly (westward) in summer and westerly (eastward) in winter. In August the winds would be easterlies, i.e. westward winds, which supports your hypothesis. The zonal wind reversal in the middle atmosphere typically occurs in September. This information can be found in a standard text book on atmospheric dynamics.

**[R1,7]   We have revised the manuscript at line 219 such that it now reads:**

> **"...aerosol plume must have been transported in the stratosphere. An easterly stratospheric wind direction at around 40º N in July is also supported by zonal mean wind fields derived from twentieth century data (Randel, 2003)."**

Equation (1): what are the limitations of this equation? What assumptions is it based on? There must be limits to the parameter ranges in which the equation is applicable, e.g. if Delta M ist very small, tau may be negative, which does not make sense.

**[R1,8] Equation [1] is derived by Stothers (1984a, 1984b). The basic principle on which it is founded is that the optical path length through the atmosphere ('air mass') varies with zenith angle (z) as sec z (see, for example, Schaefer 1993). This simple relationship is no longer true at high zenith angles where, for example, atmospheric refraction effects become important (Stothers 1984a, Schaefer 1993). Accordingly, we delimit the use of equation [1] to values of solar elevation angle ($\alpha$) $> 15$º. Thus $\sin \alpha > 0.26$ and, yes, you are correct, for a physically realistic $\tau > 0$, in theory $\Delta M > 0.77$. In the context of our paper, however, the very lowest discernible change in solar magnitude that we could consider analytically is much larger, $\Delta M = 3.4$ (manuscript line 238). We do take the point but trust that the combination of our references to Stothers (1984a, 1984b) and our delimited parameter ranges ($\alpha > 15$º and $\Delta M > 3.4$) should provide adequate assurance regarding the valid use of equation [1]. We have nevertheless revised the manuscript at line 231 to read that:**

> **"Atmospheric refraction can be neglected for $\alpha > 15^o$ (Stothers, 1984b; Schaefer, 1993)."**

Line 242: "These three observational phases"

It's not entirely clear which three phases you mean here.

**[R1,9] In the interests of brevity, we had compressed the discussion in sect. 3.6. but we appreciate that it might have lacked a degree of clarity as a result. We have therefore revised the manuscript at line 241 to add the following sentence:**

> **"For suitable aerosol optical depth values ($\tau$), three observational phases may therefore be distinguished: at higher solar elevations, a sun of normal or near-normal appearance (for ΔM < 3.4 and likely for some part of the range ΔM = 3.4 to ΔM = 12); at lower solar elevations, a pale sun able to be viewed with the naked eye (for the remaining part of the range ΔM = 3.4 to ΔM = 12); and at lower solar elevations still, a blue[(+)] sun able to be viewed with the naked eye (between ΔM = 12.5 and ΔM = 16.5). "**

Line 258: "and using the corresponding magnitude range (either 8 < \Delta M < 12 or 12.5 < \Delta M < 16.5)"

Which range is used in which case?

**[R1, 10] Likewise we have revised the manuscript at lines 249 *et seq.* to read as follows:**

> **"Nine of the sources (Appendix A) report the local time at which a blue[(+)] sun was observed. The corresponding solar elevation angle (α) can be recovered from this local time, for example, using the National Oceanic & Atmospheric Administration (NOAA) Solar Calculator (available at: https://gml.noaa.gov/grad/solcalc/) (Appendix A, col. 6). Using this solar elevation angle (α) and the range of reduction in solar magnitude associated with a blue[(+)] sun observation (ΔM = 12.5 to ΔM = 16.5), expression [1] therefore yields a corresponding range of instantaneous aerosol optical depth values ($\tau$) in each case (Fig. 8).**
>
> **Five of the sources (Appendix A) report the local time at which the sun was observed with the naked eye after having been observably blue[(+)] in the morning or before becoming observably blue[(+)] in the afternoon. The qualitative descriptions of the appearance of the sun in these latter reports, for example, as a 'crystal globe' (source [A8]) or as 'moon-like' (source [A10]), suggest the upper end of the 3.4 < ΔM < 12 range in magnitude reduction, *i.e.* 8 < ΔM < 12. Likewise, recovering solar elevation angle (α) from local time and using this solar elevation angle (α) with the range of reduction in solar magnitude associated with a naked eye sun observation (ΔM = 8 to ΔM = 12), expression [1] yields a corresponding range of instantaneous aerosol optical depth values ($\tau$) in each case (Fig. 8)."**

**and accordingly:**

> **Figure 8. Estimated instantaneous aerosol optical depth ranges. Those marked with a circle represent ranges derived from observations of a blue[(+)] sun whereas those marked with a square represent ranges derived from observations of a naked eye sun (either after having been observably blue[(+)] in the morning or before becoming observably blue[(+)] in the afternoon). (See sect. 3.6).**

Figure 8 and related explanations in the text: I don't fully understand, how the optical depths and their error bars are estimated. You use equation 1 and obtain information on the solar zenith (or elevation angles), but how is \Delta M determined for each case? This seems highly arbitrary and should introduce significant uncertainties.

**[R1,11] Please see [R1, 10].**

Please also explain, how the error bars in Fig. 8 were determined.

**[R1, 12] Please see [R1, 10].**

Line 289: "extinction co-efficient Q"

Q needs to be dimension-less for equation (2) to yield the correct units. This already implies that Q is the "extinction efficiency", not an "extinction co-efficient". Coefficients typically have units of 1/length, X-sections of length^2 and the Mie efficiencies (scattering or extinction) are dimension-less.

**[R1, 13] We have revised the manuscript at line 289 (and correspondingly in Table 2) to read 'extinction efficiency'.**

Line 340: "The eruption has been assigned a VEI of 3, which is associated with a total volume of erupted tephra of the order of 0.1 km^3."

This amount of erupted tephra is certainly probably associated with large uncertainties, right? I suggest mentioning this.

**[R1, 14] Estimating the amount of tephra erupted by a particular eruption can indeed be associated with large uncertainties. However, we say that this eruption has been assigned a VEI of 3 (which it has) and that a VEI of 3 is associated with a volume of erupted tephra of the order of 0.1 $km^3$ (which it is). We do take the point but we trust that 'of the order of' should be sufficient to flag this uncertainty *i.e.* it is unlikely to have been 0.01 $km^3$ or 1 $km^3$ but somewhere in-between.**

Line 364: "This hypothesis"

Which hypothesis do you mean here? That Ferdinandea was the source of the aerosol leading to the blue sun observations? Or your hypothesis on the additional release of sulfur?

**[R1, 15] We have revised the manuscript to refer to hypothesis 'H1' (relating to the release of sulphur from sedimentary deposits) at lines 348, 364, 555 and 562 and hypothesis 'H2' (relating to the sulphur enriched aerosol having reached the stratosphere) at lines 376, 390, 421, 436, 557 and 562.**

Line 421: „This hypothesis"

Again, which hypothesis?

**[R1, 16] Please see [R1, 15]**

Line 441 – 443: As mentioned above, the zonal winds in the stratosphere in July/August will most likely have been easterly/westward, in good agreement with your hypothesis.

Citations (occurs many times in the paper): "et al" -> "et al."

**[R1, 17] We have revised the manuscript to read 'et al.' where-ever the term is used.**

---

## Author Comment (AC2)

**Authors' response to Reviewer #2: Dr Fred Prata**

This paper presents the hypothesis that an important climate forcing event in the 19th century and seen in the records of Greenland ice cores, was due to the eruption of an undersea volcano (Ferdinandea) off the coast of Sicily that occurred in 1831.

This is an innovative, interesting and scholarly research paper and it should be published. It is innovative in its use of "eye-witness" observations of certain atmospheric optical phenomena associated with volcanic aerosols to estimate size, composition, transport and even the amount of S in a little known volcanic eruption.  It is interesting because it reveals new information on processes affecting the injection of aerosols into the stratosphere as well as perhaps solving the mystery of a sulphate signal in the Greenland ice core record thought to be from an 1831 tropical eruption.  It is scholarly (there are > 120 references) as it involves collection and research of historical documents and records in several different languages, as well as covering topics in volcanology, atmospheric optical physics and dynamics and even some radiative transfer.

**[R2,1]   Thank you, Dr Prata. We are grateful for your helpful and detailed comments.**

**Specific Comments**

The hypothesis presented relies on three key points:

1.   That a small (VEI 3) volcanic eruption could inject aerosols into the stratosphere.
2.   That the event injected sufficient SO2 to register a signal in the Greenland ice core.
3.   That the aerosols were transported westwards in agreement with the (indirect) observations presented.

I cannot comment on point 2 as I am not sufficiently expert to assist the discussion, but I can provide some thoughts on points 1 and 3.

Point 1. The idea that an undersea, relatively minor eruption could produce stratospheric aerosols seems surprising.  However, as the authors note, the eruption of Krakatau in December 2018 did just that.  More recently, the 12 August 2021 eruption of the undersea volcano Fukutoku-Okanoba, off Japan generated a column that reached up to 18 km (in the stratosphere).  These eruptions were monitored by modern satellite instruments and provide unambiguous evidence of stratospheric injection.  The 1963 eruption of Surtsey was reported to have generated a column up >9000 m which is stratospheric at the high latitude of Surtsey.  Thus there is evidence that near sea-surface volcanic eruptions can generate tall columns, even if the amount of solid-rock material is not large.  The mechanism for this is thought to be due to a combination of tropospheric convective instability and the interaction of hot material with sea water, generating additional  convective available potential energy (CAPE) which drives the buoyancy of the column.  I think the authors should perhaps describe this in more detail as they must convince readers that a small VEI eruption can generate a stratospheric aerosol – crucial to their hypothesis.

**[R2,2] It is indeed important to establish that relatively modest phreatomagmatic eruptions at sea-level can inject aerosol into the stratosphere but, as you say, the 2018 Anak Krakatau eruption and the 2021 Fukutoku-Okanoba eruption both provide 'unambiguous evidence' that they do. In this paper, our aim can therefore be narrowed to making a plausible case that the same could have happened in the different context of**

**the 1831 eruption. We trust that our discussion in manuscript lines 374 - 434 is sufficient to make this plausible case and that, in the ordinary way, our reference to Prata *et al.* (2018) is sufficient in terms of detailing at least key features of the CAPE mechanism. In terms of a more detailed application of the CAPE mechanism in the context of the 1831 eruption (*i.e.* to test our hypothesis on this point), however, that will involve a set of further considerations including choices of assumptions and data sets which we do think can only be satisfactorily addressed elsewhere.**

Point 3. The authors estimate the zonal transport of the aerosol to be ~20 m/s westwards. This is quite fast. There are no detailed vertical wind profiles available for 1831 but there are modern climatologies and also good knowledge of the atmospheric circulation based on solid meteorological foundations. Eruptions from Mt Etna (a volcano close to Empedocles/Ferdinandea) generate volcanic plumes that predominantly move eastwards and sometimes northwards and southwards, but rarely westwards. This is because these eruptions are mostly tropospheric where the winds in summer are from the west. Modern climatologies of the vertical zonal winds suggest that up to the tropopause at 40 N during the NH summer winds are from the west. Above the tropopause the winds gradually shift to the east in accordance with thermal winds caused by the vertical gradient of temperature. As the sign of the gradient changes from negative to positive the winds change from eastwards to westwards. But, according to modern zonal wind climatologies for midlatitude NH summers, the magnitude of the westward winds is much less than 20 m/s. At 70 hPa (~18 km) it is generally <10 m/s and does not reach ~20 m/s until 10 hPa (~30 km). It is possible that in August 1831 the stratospheric winds were anomalously high, but that would be somewhat speculative and convenient. I wonder then whether the authors should re-evaluate the speed of transport or perhaps put an error range on their estimate to allow for this inconsistency.

**[R2,3] As yet, our conclusion as to the 20 ms$^{-1}$ aerosol transport velocity is admittedly based on relatively sparse data (sect. 3.4, Figure 7, sect. 4.2 and Figure 11). Assuming even observational errors of ± 12 hours (*i.e.* morning *vs* evening observation or *vice versa),* will not materially change it. We certainly take the point, though, so to flag the potential for an apparent inconsistency or that the aerosol could have been injected at a greater altitude, we have revised the manuscript at line 441 to include the following paragraph:**

> **"Zonal mean wind fields derived from twentieth century data suggest that easterly wind velocity (40° N, July) would not be expected to exceed about 10 ms$^{-1}$ below an altitude of approximately 20 km (30 hPa) in the low stratosphere and that a velocity of about 20 ms$^{-1}$ would only be expected to be reached at an altitude of approximately 35 km (8 hPa) in the mid-stratosphere (Randel, 2003). A particular focus for this hypothesis testing will therefore examine whether the aerosol is likely to have reached only the low stratosphere, in which case the easterly aerosol transport velocity estimated in section 3.4 (20 ms$^{-1}$) would be inconsistent with the easterly wind velocity suggested for the lower stratosphere by the zonal mean wind field data (10 ms$^{-1}$) by a factor of two, or whether it could have reached the mid-stratosphere."**

None of these points are serious enough for me to suggest a revision is needed and I am happy to recommend publication subject to technical corrections. Indeed I found the paper so interesting that I wondered whether the authors had exhausted all observations, such as reports from ships logs (perhaps too scant?),

**[R2,4] Thank you. In fact, the literature search presented in this paper already extended over several years. There are no doubt many further observations that remain to be found, though, and we certainly encourage the testing and refinement of our reconstruction through searches for further such observations (manuscript lines 560 – 561). Yes, although we have not yet had the time to review them in earnest in this case (although see, for example, source B1), ships logs are potentially an excellent further source of high-quality observations, as demonstrated by, for example, the RECLAIM project (Wilkinson et al.).**

were pumice rafts observed?

**[R2,5] Although they were not a primary focus of the literature search, yes, pumice rafts were reported in the vicinity of the eruption.**

or could the possibility that the aerosols reached the upper stratosphere be sustained (where there are stronger winds), in which case one might expect observations of PSCs during the NH winter.  I believe the earliest documented evidence for PSCs dates back only to the 1870s.

**[R2,6] Please see [R2, 3]. We have certainly been interested in this possibility. To that end, one of us (CG) has also collated a body of observations of unusual twilight phenomena which took place from August 1831 for reporting and analysis elsewhere. To take the point more immediately, though, we have revised the manuscript at line 570 to raise the issue for future work in the following way:**

> **"Analysis of reported observations of unusual atmospheric optical phenomena both in 1831 and in 1883 may support further investigation in a number of additional directions...Further, the duration of reported twilight glow observations in 1883 were used to constrain the altitude of the aerosol responsible (Symons et al., 1888; Meinel & Meinel, 1991). In the context of testing hypothesis H2, an analysis of the duration of the reported twilight glow observations mentioned in sect. 4.2, as well as of a supplementary collected body of contemporary observations of twilight glows (and other unusual twilight phenomena) should provide independent evidence as to the altitude reached by the aerosol responsible in 1831."**

Minor comments

These are really quite minor.

1. I don't think it is necessary to have a full-stop after the longitude/latitude directions (e.g. N. should be N)

**[R2,7] We have removed full-stops from the cardinal directions in the revised manuscript.**

2. Extinction coefficient and extinction efficiency factor are both acceptable but just use one. I think extinction efficiency is most commonly used.

**[R2,8] The term 'extinction efficiency' is used in the revised manuscript. Please see [R1, 13].**

3. I suppose that sometimes observations were hampered by cloudiness. One cannot expect a particularly complete set of observations of the Sun so I think some inconsistencies could be explained by lack of visibility.

**[R2, 9] Source [A19] reports that the overcast sky was of a 'threatening' dark bluish colour but, yes, it is not clear whether thicker cloud layers could elsewhere have made observation of the phenomenon impossible.**

4. Were there any lunar observations? (The Moon was 43% illuminated and 23 days old on 14 August 1831).

**[R2,10] Yes, there were several observations of a blue[(+)] moon reported in the sources collected in Appendix A but in the interests of brevity we did not include them.**

5. I noticed in Table 1 A22 that there was mention of a "black dot" observed on the Sun. Could this be a sunspot? If so,it adds weight to the drastic diminution of light from the Sun, as observing sunspots requires considerable rejection of sunlight.  There are > 200 observations of sunspots made in 1831 (see: https://onlinelibrary.wiley.com/doi/abs/10.1002/asna.201111601) some with notes.  These records are held at the archives of the Royal Astronomical Society in London, so it might be possible for one of the authors to view them and investigate whether anything unusual was noted on solar observations during August 1831.

**[R2,11] Yes, there were several observations of a sunspot (or, depending on the visual acuity of the observer, a group of sunspots) reported in the sources collected in Appendix A but in the interests of brevity we did not include them. The presence of a large sunspot tends to increase the likelihood that observers will notice the reduced magnitude (manuscript section 3.6: between ΔM = 3.4 and ΔM = 12) of a naked eye sun. Given, perhaps, a similar interest in (naked eye) sunspots as the reviewer, one of us (CG) has collated these and other contemporary observations of the naked eye sunspots in August 1831 for reporting and analysis elsewhere (and had indeed paid an initial visit to the RAS (!) to look over Schwabe's observations with a view to using them as a baseline set against which to judge the naked eye observations).**

---

## Author Comment (AC3)

Authors' response to Anonymous Reviewer #3

This is an interesting paper that comes up with a new hypothesis concerning the now-unknown (previously attributed to Babuyan Claro) 1831 eruption. The volcanic signature seen in ice cores is attributed to an eruption in the vicinity of Sicily, which according to its assigned VEI should not have had a big impact on the stratosphere. However, associated cross-tropopause exchange could have injected sulfur from this plume into the stratosphere. The authors present a detailed study of sightings of coloured sun, which beautifully align. They are consistent with a plume that travelled around the globe form East to West in the lower stratosphere. The paper is very interesting and thought-provoking. I think this is the sort of papers that play an important role in the discussions in the community. Combining historical documentary sources and scholarly expertise with scientific reasoning is interesting. This is fascinating and should be published with some minor revisions.

**[R3,1]   Thank you. We are grateful for your helpful and detailed comments.**

However, at some instances this reasoning should be strengthened, as is outlined below.

- The time (and space) between the eruption and the first observations is very small. Sightings of coloured suns were made immediately (days) after the eruption and near the location of the eruption. This sound plausible, but is this to be expected? We read that a radius of ca. 0.5 um is required for the volcanic aerosols to have this effect. I am not a microphysicist, but in the case of tropical eruptions, sulfate aerosols, forming from the gas phase, need some time to grow to that size. This could take 2-3 months. During that time the cloud would long have circled the globe. Perhaps this is different in this case, but I would appreciate a discussion of the aerosol formation and growth process.

**[R3,2] Yes, this is indeed an interesting point but one which we had decided could not be satisfactorily addressed in this paper. It is also raised by reviewer # 1. Rather than what would be a lengthy and speculative discussion at this stage, we have therefore revised the manuscript at line 570 to raise the issue for future work in the following way:**

> **"Analysis of reported observations of unusual atmospheric optical phenomena both in 1831 and in 1883 may also support further investigation in a number of additional directions. For example, the first observation of a blue[(+)] sun in the equivalent connected sequence following the onset of the most explosive phase of the 1883 Krakatau eruption occurred of the order of one day later (Symons *et al.*, 1888). This suggests the rapid formation of a stratospheric aerosol whose size distribution is dominated by particles with a radius of the order of 0.5 μm (Table 2). The close co-incidence between the substantial increase in the intensity of phreatomagmatic activity during the Ferdinandea eruption between 6 and 11 August 1831 and the first observation of a blue[(+)] sun in the connected sequence on 8 August 1831 (sect. 4.1) suggests that a similarly rapid stratospheric aerosol formation occurred in that case too. Given that, for example, the stratospheric aerosol produced by the 1991 Pinatubo eruption took several months to grow to a typical size between 0.3 and 0.5 μm (Self *et al.*, 1993), it would be interesting to consider the nature of the atypical microphysical processes that could be involved in these two rare cases. Further..."**

Is there a role of ash or is ash too large anyway for that?

**[R3,3] Phreatomagmatic eruptions do, of course, produce fine ash. However, it is not possible to infer the composition of the aerosol particles from the blue[(+)] sun observations so we cannot say what role it might have played either in terms of optical properties or aerosol particle nucleation. The vast majority of phreatomagmatic eruptions do not produce blue[(+)] sun aerosols of the type produced in 1831 and 1883.**

- How likely is it that "tropospheric" plumes at these latitudes reach the stratosphere due to trop-strat exchange? There are examples from other locations, but is there also evidence from this region, e.g., from Etna eruptions?

**[R3,4]    We are not aware of any significant quantities of aerosol being injected into the stratosphere from eruptions at Etna whether by direct injection or other troposphere / stratosphere exchange mechanisms.  The only significant sulphate peak identified in Greenland ice-cores which has so far been attributed to an eruption in the region appears to be the 79 AD / CE eruption of Vesuvius (Sigl et al., 2013), which is assigned a VEI of 5 (associated with definite (direct) stratospheric injection) (Global Volcanism Program, 2013).**

20 m/s is superfast. Perhaps you can provide some context for this?

**[R3,5]   Please see our response 3 to Reviewer #2.**

You mention 20CRv3, but then do not show it. I agree that for this region and time period, this data set might not show much, but a plot would still be nice (the data go back to 1806 on NCAR's website).

**[R3,6] We are aware of data available for the 1830s for both 20CR V2 and V3, however data pre-1836 is not yet part of the official release. We are currently working on how we might be able to use 20CR data to investigate hypothesis H2 but prefer to leave this to 'future research'.**